# Robust Multi-Stage Nonlinear Model Predictive Control Using Sigma Points

**Sakthi Thangavel [1],\* , Radoslav Paulen [2]  and Sebastian Engell [1]**

1   Process Dynamics and Operations Group, Department of Chemical and Biochemical Engineering, Technische Universität Dortmund, Emil-Figge-Strasse 70, 44227 Dortmund, Germany; sebastian.engell@tu-dortmund.de
2   Faculty of Chemical and Food Technology, Slovak University of Technology in Bratislava, 81237 Bratislava, Slovakia; radoslav.paulen@stuba.sk
\*   Correspondence: sakthi.thangavel@tu-dortmund.de; Tel.: +49-231-755-5341

**Abstract:** We address the question of how to reduce the inevitable loss of performance that is incurred by robust multi-stage NMPC due to the lack of knowledge compared to the case where the exact plant model (no uncertainty) is available. Multi-stage NMPC in the usual setting over-approximates a continuous parametric uncertainty set by a box and includes the corners of the box and the center point into the scenario tree. If the uncertainty set is not a box, this augments the uncertainty set and results in a performance loss. In this paper, we propose to mitigate this problem by two different approaches where the scenario tree of the multi-stage NMPC is built using sigma points. The chosen sigma points help to capture the true mean and covariance of the uncertainty set more precisely. The first method computes a box over-approximation of the reachable set of the system states whereas the second method computes a box over-approximation of the reachable set of the constraint function using the unscented transformation. The advantages of the proposed schemes over the traditional multi-stage NMPC are demonstrated using simulation studies of a simple semi-batch reactor and a more complex industrial semi-batch polymerization reactor benchmark example.

**Keywords:** adaptive control; economic model predictive control; multi-stage decision making; robust model predictive control; parameter estimation; parameter uncertainty; unscented transformation

## 1. Introduction

The process industries strive to cut down their operational costs while adhering to strict quality, safety, and environmental specifications. This results in an increased interest in using optimization-based control strategies to control the plants. Among the different optimization-based controllers, model predictive control (MPC) is most widely used because of its ability to handle multivariate systems with constraints efficiently [1,2].

MPC uses a plant model to predict the future evolution of the plant over a certain period, known as the prediction horizon, and optimizes the future control moves with respect to a performance criterion (tracking or economic). Only the control input that was obtained for the first time step by solving the optimization problem is applied to the plant. At the next time step, the MPC optimization problem is reinitialized with the plant measurements and resolved. This is commonly known as the moving horizon strategy [3,4].

The performance of any model-based controller depends strongly on the accuracy of the model used. Often there exists a discrepancy between the true plant dynamics and the model predictions due to external disturbances, inaccurate model parameters or structural plant–model mismatch. This may lead to constraint violations or performance degradation when the plant is controlled using the MPC

controller with a nominal model. Robust MPC strategies take the uncertainties into account when computing the control moves [5–10]. The most prominent ones are min-max MPC, tube-based MPC, and multi-stage MPC.

Min-max MPC extends the idea behind min-max optimal control [11] to the MPC framework. Min-max MPC solves the MPC optimization problem for the worst-case realization of uncertainty [6]. The initial work on min-max MPC does not take into account the presence of future feedback information, hence it is rather conservative and may easily result in infeasible optimization problems [7]. A closed-loop min-max MPC formulation was presented in [12,13] where the future availability of measurement information and that the future control moves can be adapted depending on the realization of the uncertainty is taken into account. The closed-loop formulation requires solving an optimization over all possible feedback policies and results in a computationally very hard problem. It can be simplified by assuming a fixed control structure along the prediction horizon [12] at the expense of reduced performance.

Tube-based MPC [8,14] computes robust control actions using set-theoretic methods. It consists of two controllers, the nominal controller and the ancillary controller. The nominal controller considers tightened constraints and uses the nominal model of the plant to predict the trajectory of the system in the absence of uncertainty. The ancillary controller keeps the true plant dynamics in the neighborhood of the nominal trajectory in the presence of uncertainty such that the original constraints are satisfied. Several variants of the tube-based MPC are proposed in the literature with various levels of computational complexity and conservatism [15–19]. The main differences among these approaches lie in the computation of the uncertainty region around the nominal trajectory and the chosen ancillary controller. A variant of tube-based nonlinear MPC without the ancillary controller has been presented in [20]. The controller constructs a robust forward reachability tube that encloses the true plant trajectories in the presence of feedback actions using min-max differential inequalities.

Multi-stage MPC models the uncertainty by a tree of discrete scenarios [10]. It leads to an open-loop formulation of the optimal closed-loop control problem for the uncertainties that are included in the scenario tree and is less conservative when compared to other robust NMPC approaches [21]. For a nonlinear system, multi-stage NMPC rigorously guarantees constraint satisfaction for the uncertainties that are explicitly considered in the scenario tree. In the presence of continuous-valued uncertainty, representing all possible values of the uncertainty is infeasible. Usually, a good trade-off between robustness and computational cost is achieved by generating the scenario tree for all combinations of the minimum, nominal and maximum values of the uncertain parameters or disturbances [21,22]. This results in a box over-approximation of the true uncertainty region.

The application of any robust control mechanism inevitably results in conservatism and a loss of performance compared to the case when perfect information about the system is available. The performance loss of the robust schemes is related to the amount of uncertainty that is considered. If the parameters are estimated from data, the uncertainty can often be represented by ellipsoids that define the confidence region of the uncertain parameters [23,24]. Over-approximating an ellipsoidal confidence region by a box when generating the scenario tree of multi-stage NMPC then leads to an additional loss of performance.

The scenario tree of multi-stage NMPC can be generated using sigma points such that it more tightly approximates the uncertainty set, resulting in better performance. The sigma points capture the mean and the covariance of the uncertainty set [25,26]. In the unscented Kalman Filter [25,27,28], the sigma points in the state space are propagated to get an approximation of the distribution of the a-priori state estimate. Here we propagate the uncertain parameters which usually are less than the dimension of the state space. Then a box over-approximation of the reachable set can be computed from the propagated sigma points.

Several other papers also develop robust nonlinear model predictive control schemes using the unscented transformation principle [29–33]. Robust NMPC using the unscented transformation computes the statistical moments of the process trajectory in the presence of uncertain initial

conditions and parameters using the unscented transformation [30]. This approach was extended to stochastic model predictive control with chance constraints in [31] and its advantages were shown for an automotive emergency braking system with collision avoidance in [32]. These approaches are open-loop approaches because they do not take into account the presence of future feedback information. A closed-loop formulation of robust MPC using the unscented transformation can be obtained by embedding a state estimation scheme based on the unscented transformation (e.g., the unscented Kalman filter) into the model predictive control optimization problem in order to achieve robustness against the estimation error [29,34]. Similar work was carried out in this direction by [33,35,36]. These papers mainly differ in the way in which the unscented Kalman filter equations are considered in the NMPC optimization problem. Since the complete unscented Kalman filter equations are embedded into the NMPC optimization problem, these approaches are in general computationally demanding.

In this paper, we propose two novel computationally efficient closed-loop robust multi-stage NMPC strategies that are based upon using the sigma points and unscented transformation principles. The scenario tree of the proposed schemes is generated using sigma points. The sigma points tightly approximate the uncertainty set in contrast to standard multi-stage NMPC where the uncertainty set is over-approximated by a box [21] and result in a better performance.

In the first approach, we propagate the sigma points along the prediction horizon to compute the state mean and the state covariance matrix using the unscented transformation. A box over-approximation of the reachable set of states is computed along the prediction horizon using the state mean and the scaled covariance matrix. This takes into account that the dimension of the uncertain parameters is usually smaller than the dimension of the state space as mentioned before. The objective and constraint function of the NMPC are evaluated for the mean and for the vertices of the box over-approximation of the reachable set of states. The initial work done in this direction showed promising results [37,38]. In the second approach, we compute the mean and the covariance matrix for the vector of constraint functions evaluated at the state predictions, and a box over-approximation of the reachable set of the constraint functions is obtained similar to the first approach. This approximates the reachable set of the constraint functions tighter when compared to the first approach and results in a better performance.

The proposed schemes take into account the effect of future control moves on the over-approximated reachable sets of the model and provide a closed-loop formulation as recourse (dependency of future control inputs on future information) is included. This results in a better performance when compared to open-loop approaches [30,31]. The proposed approaches require propagation of sigma points and only a part of the unscented Kalman filter equations have to be embedded into the NMPC optimization problem when compared to some of the existing robust NMPC schemes using the unscented transformation [29,30,33], hence the proposed approaches are computationally less demanding. Full state measurements are assumed, but the schemes can be combined with existing output feedback schemes [39–41] in a straightforward manner to obtain robustness against estimation errors.

The closed-loop performance of the controller can further be improved using the measurements obtained from the controlled system. We use the measurements to enhance the knowledge about the system via parameter estimation, e.g., in the least-squares sense, and thus to reduce the range of uncertainty. This is usually referred to as adaptive robust control [42–45]. An adaptive variant of the proposed robust multi-stage NMPC scheme using sigma points is presented also in this paper, extending the initial work in this direction in [38]. We compute an optimal ellipsoidal over-approximation of the intersection of the initial confidence region and the confidence regions obtained from the measurements and update the scenario tree of the adaptive multi-stage NMPC whenever a new measurement information from the plant becomes available. This results in a better performance if the observed measurements provide more information about the uncertain parameters when compared to the initial confidence region. The performance of the different robust

NMPC controllers that are proposed in this paper is compared for a simple semi-batch reactor benchmark example and for a more complex industrial semi-batch polymerization reactor case study. Extensive simulation studies on the simple case study are performed to analyze the influence of the tuning parameters on the performance of the different robust multi-stage NMPC schemes.

The remainder of the paper is organized as follows. The problem statement and the unscented transformation are explained in Section 2. Standard multi-stage NMPC [10,21], and the proposed multi-stage NMPC using sigma points are described in Section 3. The adaptive variants of the NMPC schemes are presented in Section 4. The case studies along with the results are discussed in Section 5. Finally, the paper is concluded in Section 6.

## 2. Preliminaries

The model of the plant is assumed to be known, yet some of the model parameters are unknown. The true values of the uncertain parameters are assumed to be contained in a set. The model equations are given by Equations (1) and (2):

$$x_{k+1} = f(x_k, u_k, d),\tag{1}$$

where $x \in \mathbb{R}^{n_x}$ and $u \in \mathbb{R}^{n_u}$ represent the state and control variables. The vector $d \in \mathbb{D}(d_0, P_0) \subset \mathbb{R}^{n_d}$ represents the uncertain model parameters that are contained in an ellipsoidal set $\mathbb{D}(d_0, P_0)$. The ellipsoidal set could represent the confidence region of an estimation of the uncertain parameters and is described by

$$\mathbb{D}(d_0, P_0) := \{d \in \mathbb{R}^{n_d} | (d - d_0)^T P_0^{-1} (d - d_0) \leq 1\},\tag{2}$$

where $d_0 \in \mathbb{R}^{n_d}$ represents the nominal values of the uncertain parameters and acts as the center of the confidence region, and $P_0 \in \mathbb{R}^{n_d \times n_d}$ represents the parameter covariance matrix and describes the shape of the confidence region. The set is assumed to form a finite support of the probability distribution of the model parameters. All the states are considered to be measured directly from the plant. $x_s^m \in \mathbb{R}^{n_x}$ gives the plant measurement obtained at $s^{\text{th}}$ sample, where $s = \frac{t}{t_s}$, $t$ represents the current time and $t_s$ represents the sampling time of the plant. The state measurements are assumed to be corrupted by uncorrelated white Gaussian noise with a covariance matrix $\Sigma = \text{diag}^2(\sigma)$ with $\sigma \in \mathbb{R}^{n_x}$.

The unscented transformation is used to compute the statistics of variables which undergo a nonlinear transformation [26,46,47]. The principle behind the unscented transformation is illustrated in Figure 1, where $n_d = 2$ and the uncertain parameters undergo a nonlinear transformation by an arbitrary function $n : \mathbb{R}^{2 \times 2} \to \mathbb{R}^2$. The pink shaded region in Figure 1a represents the confidence region of the uncertain parameters and in Figure 1b it represents the image set of the nonlinear function $n$ (i.e., $n(d)$, $\forall d \in \mathbb{D}(d_0, P_0)$). $2n_d + 1$ points known as the sigma points (marked with red squares in Figure 1a) are chosen such that they capture the true mean and the covariance of the uncertain parameter set $\mathbb{D}(\cdot)$ and are given as Equations (3) and (4):

$$\mathbb{S}(d_0, P_0) = d_0 \cup \left( \bigcup_{i=1}^{n_d} d_0 - P_{0,[i,\star]}^{\frac{1}{2},T} \right) \cup \left( \bigcup_{i=1}^{n_d} d_0 + P_{0,[i,\star]}^{\frac{1}{2},T} \right),\tag{3}$$

where $P_{0,[i,\star]}^{\frac{1}{2},T}$ gives the transpose of the $i^{\text{th}}$ row vector of the matrix square root of $P_0 \in \mathbb{R}^{n_d \times n_d}$ obtained using Cholesky decomposition. The sigma points are propagated through the nonlinear function $n(\cdot)$ to compute the mean and covariance matrix of the transformed points that result from the nonlinear transformation.

$$\chi^i = n(d^i), \qquad\qquad \forall i \in I_{sp}, d^i \in \mathbb{S}(d_0, P_0),\tag{4a}$$

$$\boldsymbol{\mathcal{x}}_m = \sum_{i=1}^{2n_d+1} \boldsymbol{v}_{[i]} \boldsymbol{\mathcal{x}}^i, \quad \sum_{i=1}^{2n_d+1} \boldsymbol{v}_{[i]} = 1, \tag{4b}$$

$$\boldsymbol{\mathcal{X}}_c = \kappa^2 \sum_{i=1}^{2n_d+1} \boldsymbol{v}_{[i]} \left( \boldsymbol{\mathcal{x}}^i - \boldsymbol{\mathcal{x}}_m \right) \left( \boldsymbol{\mathcal{x}}^i - \boldsymbol{\mathcal{x}}_m \right)^T, \tag{4c}$$

where $I_{sp} := \{1, \cdots, 2n_d + 1\}$, $\boldsymbol{d}^i$ represents the $i^{\text{th}}$ element of the set $\mathbb{S}(\boldsymbol{d}_0, \boldsymbol{P}_0)$ and $\boldsymbol{v} \in \mathbb{R}^{2n_d+1}$ are the sigma point weights, so $\boldsymbol{v}_{[i]}$ represents the weight associated with the sigma point $\boldsymbol{d}^i$. The sigma points are propagated using the nonlinear function $\boldsymbol{n}(\cdot)$ using Equation (4a) as shown using red dotted lines in Figure 1. The mean ($\boldsymbol{\mathcal{x}}_m$) and the covariance matrix ($\boldsymbol{\mathcal{X}}_c$) are computed using Equations (4b) and (4c). $\kappa \in \mathbb{R}$ is the scaling factor of the covariance matrix. The ellipsoids obtained using the mean and the covariance matrix for different values of the scaling factor are shown in blue in Figure 1. The function that computes the mean and covariance matrix of a nonlinear transformation using unscented transformation can be compactly represented as $(\boldsymbol{\mathcal{x}}_m, \boldsymbol{\mathcal{X}}_c) = \mathbb{U}(\kappa, \boldsymbol{\mathcal{x}}^{[1:2n_d+1]})$, where $\boldsymbol{\mathcal{x}}^{[1:2n_d+1]}$ represents values corresponding to $2n_d + 1$ evaluations of the nonlinear function $\boldsymbol{n}(\cdot)$ (see Equation (4a)). e.g., in Figure 1, the values $\boldsymbol{\mathcal{x}}^1, \boldsymbol{\mathcal{x}}^2, \boldsymbol{\mathcal{x}}^3, \boldsymbol{\mathcal{x}}^4$, and $\boldsymbol{\mathcal{x}}^5$ are represented by $\boldsymbol{\mathcal{x}}^{[1:2n_d+1]}$.

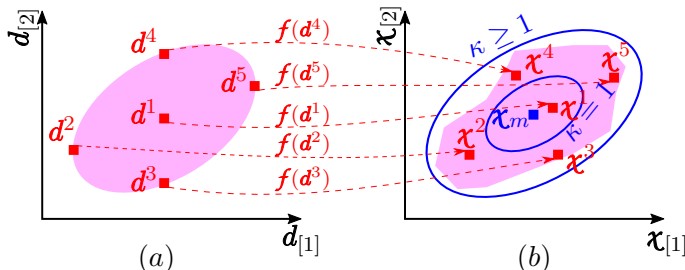

**Figure 1.** Principles of the unscented transformation. (**a**) Confidence region of the uncertain parameters (**b**) The image set of the nonlinear function $\boldsymbol{n}$.

## 3. Robust Multi-Stage NMPC

### 3.1. Standard Multi-Stage NMPC

Multi-stage NMPC [21] is a robust NMPC strategy that models the effect of the uncertainties by a tree of discrete scenarios as shown in Figure 2. Each branch of the scenario tree represents the trajectory of the system states for a given control input and a particular realization of the uncertainty that can vary at each point in the prediction horizon. Multi-stage NNPC computes control inputs while taking into account that measurement information will be available in the future and that the future control moves can be adapted accordingly, i.e., the control inputs beyond the next time step become scenario-dependent recourse variables. This allows us to solve a feedback problem as an open-loop optimization problem. The closed-loop formulation results in a significant improvement in the performance of the robust controller in comparison to open-loop schemes [7,48].

#### 3.1.1. Multi-Stage NMPC (MS NMPC)

The standard multi-stage NMPC proposed in [10,21] represents uncertainty region by a box. The scenario tree of multi-stage NMPC is usually built for all possible combinations of the minimal, maximal, and nominal values of the uncertain parameters. The formulation of standard multi-stage NMPC solved at time $t$ ($s^{\text{th}}$ sampling time) for the case where the parameter uncertainty is given by an ellipsoidal set $\mathbb{S}(\boldsymbol{d}_0, \boldsymbol{P}_0)$ reads as Equation (5):

$$\min_{\boldsymbol{x}_k^j, \boldsymbol{u}_k^j, \boldsymbol{d}^j} \sum_{k=s}^{s+N_r-1} \sum_{j=1}^{N_b^{k-s+1}} \omega_{k+1}^j l(\boldsymbol{x}_{k+1}^j, \Delta \boldsymbol{u}_k^j) + \sum_{k=s+N_r}^{s+N_p-1} \sum_{j=1}^{N_b^{N_r}} \omega_{k+1}^j l(\boldsymbol{x}_{k+1}^j, \Delta \boldsymbol{u}_k^j), \quad \forall (j, k+1) \in I_{st}, \tag{5a}$$

subject to

$$x_{k+1}^j = f(x_k^{p(j)}, u_k^j, d^{r(j)}), \qquad \forall\,(j, k+1) \in I_{st}, d^{r(j)} \in \mathcal{D}, \tag{5b}$$

$$g(x_{k+1}^j, u_k^j) \le 0\,, \qquad \forall\,(j, k+1) \in I_{st}, \tag{5c}$$

$$u_k^j = u_k^l \text{ if } x_k^{p(j)} = x_k^{p(l)}, \qquad \forall\,(j, k), (l, k) \in I_{st}, \tag{5d}$$

$$\underline{u} \le u_k^j \le \overline{u}, \qquad \forall\,(j, k) \in I_{st}, \tag{5e}$$

$$x_s^1 = x_s^m \tag{5f}$$

$$\underline{d} = d_0 - \text{diag}^{\frac{1}{2}}(P_0),\ \overline{d} = d_0 + \text{diag}^{\frac{1}{2}}(P_0), \tag{5g}$$

$$\mathcal{D} = \mathbb{C}_a(\underline{d}, d_0, \overline{d}). \tag{5h}$$

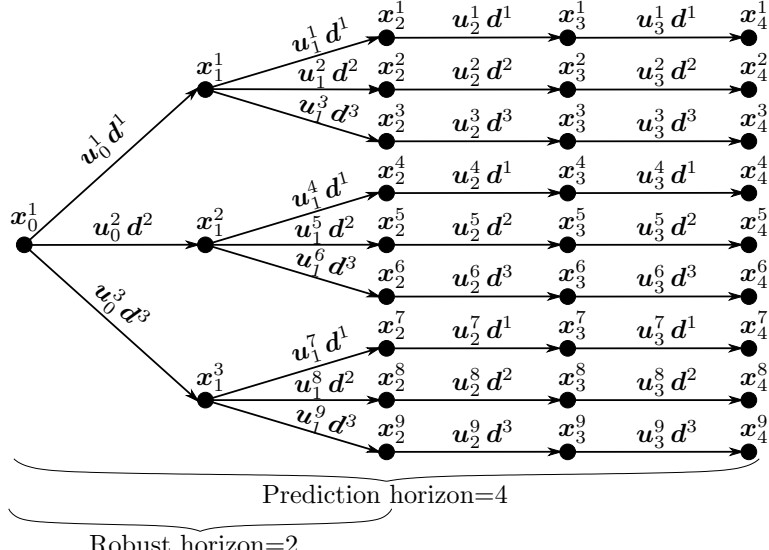

**Figure 2.** Scenario tree of the multi-stage NMPC.

The state trajectory along the branches of the scenario tree for different realizations of the uncertainty is given by Equation (5b). The state prediction $x_{k+1}^j$ at stage $k+1$ and position $j$ in the scenario tree is obtained using the parent state $x_k^{p(j)}$, the control input $u_k^j$, and the uncertainty realization $d^{r(j)} \in \mathcal{D}$. The notation is explained briefly using Figure 2 for the optimization problem solved at time $t = 0$. The root node is represented by $x_0^1$ and is initialized at current plant measurement $x_0^m$. The root node branches into the child node $x_1^1$, $x_1^2$ and $x_1^3$ depending on the applied control inputs and realization of the uncertainty. For e.g., $x_1^1$ is obtained if the uncertainty $d^1$ is realized and control move $u_0^1$ is applied at the root node $x_0^1$. The root node $x_0^1$ acts as the parent node for the child nodes $x_1^1$, $x_1^2$, and $x_1^3$. The nodes $x_0^1$, $x_1^1$, $x_2^1$, $x_3^1$, and $x_4^1$ together form a scenario of the scenario tree. $I_{st}$ denotes the set of indices $(j, k)$ that occur in a given scenario tree. The set $\mathcal{D}$ represents different realizations of the uncertain parameters considered by the standard multi-stage NMPC. The additional constraints that must be satisfied at each node in the scenario tree are given by Equation (5c). The controller cannot anticipate the future realization of the uncertainty i.e., the control decision originating from the same parent node must be the same (for e.g., in Figure 2 $u_0^1 = u_0^2 = u_0^3; u_1^1 = u_1^2 = u_1^3; \cdots$). This is enforced using the non-anticipativity constraints Equation (5d). The inputs are bounded using Equation (5e), where $\underline{u}$ and $\overline{u}$ gives the lower and upper bounds on the control inputs. The robust NMPC is reinitialized at the plant measurement obtained at the $s^{\text{th}}$ sampling time using Equation (5f). The minimum and maximum values of the uncertain parameters can be obtained using their nominal values and the parameter covariance matrix from Equation (5g), where $\text{diag}^{\frac{1}{2}} : \mathbb{R}^{n_d \times n_d} \to \mathbb{R}^{n_d}$ gives a vector that contains the square roots of the diagonal elements of the matrix. The uncertain parameter

values considered by the standard multi-stage NMPC are obtained using Equation (5h), where the operator $\mathbb{C}_a(\cdot)$ generates a set of all possible combinations of the minimum, nominal and maximal values of the uncertainty. The scenario tree grows exponentially along the prediction horizon $(N_p)$ which can be prevented by stopping the branching after a certain point in time along the prediction horizon known as the robust horizon $(N_r)$ under the assumption that the uncertainty realization remains constant after the robust horizon. The objective function of the NMPC optimization problem is given in Equation (5a), where $\Delta \boldsymbol{u}_k = \boldsymbol{u}_k - \boldsymbol{u}_{k-1}$ gives the deviation between two consecutive control moves. The first part of the objective function represents the contribution of all the nodes until the robust horizon and the second part of the objective function gives the contribution of all the nodes after the robust horizon on the objective function. $N_b$ is the number of branches at each node in the scenario tree, where $N_b = 3^{n_d}$ for standard multi-stage NMPC. $l(\cdot)$ represents the cost at each node in the scenario tree and $\omega_{k+1}^j$ is the weight associated with each node in the scenario tree and is given by Equation (6):

$$\sum_{j=1}^{N_b^{k-s+1}} \omega_{k+1}^j = 1, \quad \forall k \in I_{br}, \qquad \sum_{j=1}^{N_b^{N_r}} \omega_{k+1}^j = 1, \quad \forall k \in I_{ar}, \tag{6}$$

where $I_{br} := \{s, \cdots, s + N_r - 1\}$ and $I_{ar} := \{s + N_r, \cdots, s + N_p - 1\}$.

**Remark 1.** *The stability properties of the robust MPC have been previously analyzed in the literature using standard tracking objective functions. The first analysis was presented in [7] for linear systems where a min-max cost function is considered. Several papers have been published for linear stochastic systems and exponential stability in the mean square sense has been established [49,50]. The setting considered in this paper is more complex, as it considers nonlinear systems and a deterministic representation of the uncertainty. Initial work on the stability analysis of standard multi-stage NMPC was carried out in [51]. The consideration of different weights for the cost function at different nodes in the scenario tree (as opposed to a worst-case or probabilistic formulation) is a challenge for the stability analysis. The stability property is satisfied in [51] via continuity assumptions on the model equations. As with other robust approaches, only convergence to a neighborhood of the equilibrium point can be achieved because the different branches of the uncertainty are considered at each sampling time, even if the disturbance vanishes [52]. Convergence can be recovered by using a dual-mode approach as used in other MPC schemes [53]. A rigorous analysis of the stability properties of the multi-stage approach in the context of economic model predictive control [54,55] is still an open issue.*

3.1.2. Multi-Stage NMPC Based on the Vertex Over-Approximation (MS-VA NMPC)

In standard multi-stage NMPC all combinations of the minimum, nominal and maximum values of the uncertain parameters are chosen to approximate the uncertainty set, this results in $3^{n_d}$ branches to be considered at each node in the scenario tree. Similar performance with reduced computational effort can be achieved if we choose only the nominal parameter value along with the vertices of the box over-approximation of the uncertainty set. This results in only $2^{n_d} + 1$ branches to be considered at each node in the scenario tree. We call this multi-stage NMPC based on the vertex approximation (MS-VA). The formulation of the MS-VA NMPC optimization problem (The formulation of the entire MS-VA, MS-SB, and MS-CB NMPC optimization problems are provided in S-1 as Supplementary Material. The formulation of entire adaptive robust multi-stage NMPC optimization problem is provided in S-2 of the Supplementary Material) that is solved at time $t$ ($s^{\text{th}}$ sampling time) is similar to the optimization problem Equation (5), where Equation (5h) is replaced with Equation (7):

$$\mathcal{D} = \boldsymbol{d}_0 \cup \mathbb{C}_{vp}(\underline{\boldsymbol{d}}, \overline{\boldsymbol{d}}), \tag{7}$$

$\mathbb{C}_{vp}(\cdot)$ generates the set of all possible combinations of the lower and upper bound of the uncertainty set along with its nominal value. The objective function of the multi-stage NMPC based on the vertex

over-approximation approach is similar to the standard multi-stage NMPC Equation (5a) approach except that $N_b = 2^{n_d} + 1$.

**Remark 2.** *The uncertain parameter values chosen to build the scenario tree of the MS NMPC and MS-VA NMPC can guarantee robust constraint satisfaction in the presence of continuous-valued uncertainty only if the parametric monotonicity property of the problem constraints is satisfied (i.e., the sensitivity of the constraints with respect to the uncertain model parameters has the same sign for all parameter values within the uncertainty set) [56]. If the parametric monotonic property is not satisfied, robust constraint satisfaction is guaranteed only for the scenarios that are explicitly considered in the scenario tree of the multi-stage NMPC. In such cases, the multi-stage approach can be combined with reachability analysis as shown in [57] to obtain robust constraint satisfaction but this increases the computational complexity of the resulting robust control problem.*

*3.2. Multi-Stage NMPC Using Sigma Points*

In standard multi-stage NMPC (see Section 3.1), the uncertainty set is over-approximated even if more precise information about the uncertain parameters is available and this results in a performance loss. This will be avoided using sigma points. The sigma points ($2n_d + 1$ sample points from the boundary of the ellipsoidal uncertainty set) and their propagation can be used to approximate the box of the predicted uncertain states more tightly. In this paper, we propose two methods to make use of the propagation of the sigma points, in first method the box over-approximation of the reachable set of states is computed and in the second method a box over-approximation of the reachable set of the constraint function is employed.

3.2.1. Multi-Stage NMPC Based on the Box Over-Approximation of the Reachable Set of States (MS-SB NMPC)

The key difference between MS-VA NMPC (Section 3.1.2) and MS-SB NMPC is illustrated in Figure 3 for a system with two states, two uncertain parameters, and a given control input at time $k = 1$. Figure 3a represents the parametric space. The pink shaded region represents the ellipsoidal confidence region of the uncertain parameters given by Equation (2). Black dots (vertices of the box over-approximation of the uncertainty set) and red squares (sigma points obtained using Equation (3)) represent the parameter samples that are chosen to build the scenario tree of MS-VA NMPC and MS-SB NMPC, respectively. The nominal value of the uncertain parameter ($d^1 = d_0$) is considered in the scenario tree of both MS-VA NMPC and MS-SB NMPC.

Figure 3b,c represent the reachable set in the state space ($\mathbb{X}_1^1$) by a pink shaded region. Black circles in Figure 3b represent the state predictions obtained on the branches of the scenario tree of MS-VA NMPC (e.g., $x_1^1$ is obtained for the uncertainty realization $d^1$ when the control input $u_0^1$ is applied). Red squares in Figure 3c represent the state predictions obtained on the branches of the scenario tree of MS-SB NMPC. The inner (blue) ellipsoid represents the state covariance ellipsoid described by the state mean and the state covariance matrix (i.e., $\kappa = 1$) computed using the state predictions of MS-SB NMPC. The state covariance matrix is then enlarged such that the box over-approximation (represented by the dotted blue line) of the extended ellipsoid (represented by the outer (blue) ellipsoid) over-approximates the reachable set of states. Blue squares represent the state mean and the vertices of the box over-approximation of the reachable set for which the objective and constraint functions of MS-SB NMPC are evaluated.

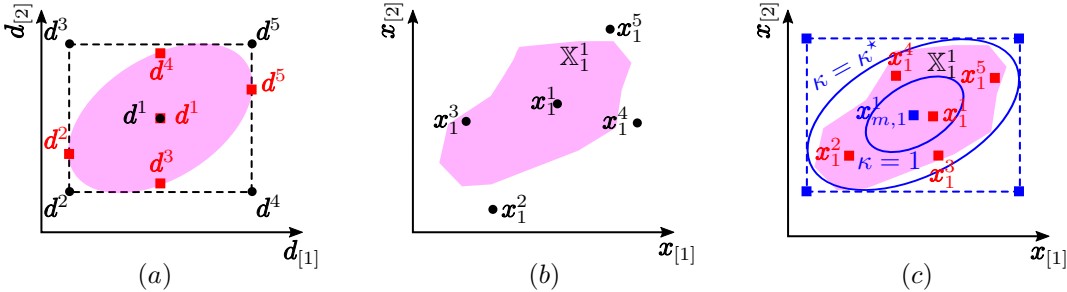

**Figure 3.** Comparison between multi-stage NMPC based on the vertex approximation and multi-stage NMPC based on the box over-approximation of the reachable states set. (**a**) Confidence region of the uncertain parameters (**b**) Predictions and the reachable set for MS-VA NMPC. (**c**) Predictions, reachable set and over-approximation of the reachable set for MS-SB NMPC.

Tuning of the Scaling Factor $\kappa$

The scaling factor $\kappa$ for the enlargement of the ellipsoidal set that results from the propagation of the sigma points can be computed such that the box over-approximation of the scaled state ellipsoids tightly over-approximates the entire reachable set of model (1) by solving $n_x$ optimization problems given below (Equation (8)), where $j \in I_x := \{1, \cdots, n_x\}$,

$$\hat{\boldsymbol{\kappa}}_{s,[j]} = \max_{\kappa, \boldsymbol{x}_0^{\mathrm{wc}}, \boldsymbol{u}_0^{\mathrm{wc}}, \boldsymbol{d}^{\mathrm{wc}} \in \mathbb{D}, \boldsymbol{d}^i \in \mathbb{S}(\boldsymbol{d}_0, \boldsymbol{P}_0)} \frac{\left| \boldsymbol{x}_{1,[j]}^{\mathrm{wc}} - \boldsymbol{x}_{m,[j]} \right|}{\sqrt{\mathbf{X}_{c,1,[j,j]}^p}}, \tag{8a}$$

subject to

$$\boldsymbol{x}_1^i = \boldsymbol{f}(\boldsymbol{x}_0^{\mathrm{wc}}, \boldsymbol{u}_0^{\mathrm{wc}}, \boldsymbol{d}^i), \qquad \forall i \in I_{sp}, \boldsymbol{d}^i \in \mathbb{S}(\boldsymbol{d}_0, \boldsymbol{P}_0), \tag{8b}$$

$$\left( \boldsymbol{x}_{m,1}, \mathbf{X}_{c,1}^p \right) = \mathbb{U}(1, \boldsymbol{x}_1^{[1:2n_d+1]}), \tag{8c}$$

$$\boldsymbol{x}_1^{\mathrm{wc}} = \boldsymbol{f}(\boldsymbol{x}_0^{\mathrm{wc}}, \boldsymbol{u}_0^{\mathrm{wc}}, \boldsymbol{d}^{\mathrm{wc}}), \tag{8d}$$

$$\boldsymbol{g}(\boldsymbol{x}_0^{\mathrm{wc}}, \boldsymbol{u}_0^{\mathrm{wc}}) \leq 0, \tag{8e}$$

$$\mathbf{X}_{c,1,[j,j]}^p > 0, \tag{8f}$$

where $\boldsymbol{v}$ is the weight associated with the sigma points and is chosen such that $\sum_{i=1}^{2n_d+1} \boldsymbol{v}_{[i]} = 1$. $\boldsymbol{d}^i$ represents the $i^{\mathrm{th}}$ uncertain parameter combination in the set of uncertain parameters obtained using $\mathbb{S}(\cdot)$. The objective function Equation (8a) gives the maximum value by which the state ellipsoid should be scaled, for the box over-approximation of the scaled state ellipsoid to enclose the reachable set with respect to state $j$. The state mean $(\boldsymbol{x}_{m,1})$ and the unscaled state covariance matrix $(\mathbf{X}_{c,1}^p)$ are obtained using Equation (8c), where $\boldsymbol{x}_1^{[1:2n_d+1]}$ represents all the model predictions obtained using the sigma points given by Equation (8b). The vector $\boldsymbol{x}_1^{\mathrm{wc}}$ denotes the prediction obtained in the whole operating region of the plant that results in the maximum value of the objective function. Vectors $\boldsymbol{x}_0^{\mathrm{wc}}$ and $\boldsymbol{u}_0^{\mathrm{wc}}$ are bounded by operating region of the plant using Equation (8e). The constraint Equation (8f) makes sure that the optimization problem becomes infeasible if the sensitivity of the states for the uncertain parameters is 0 (i.e., the uncertain parameter does not influence the state predictions). The scaling factor corresponding to the different states is given by Equation (9):

$$\hat{\boldsymbol{\kappa}}_{x,[j]}^\star = \begin{cases} 0, & \text{if Equation (8) is infeasible} \\ \hat{\boldsymbol{\kappa}}_{x,[j]}, & \text{otherwise} \end{cases}, \qquad \forall j \in I_x \tag{9}$$

where $\hat{\boldsymbol{\kappa}}_x^\star \in \mathbb{R}^{n_x}$. The scaling factor is set to 0 if the uncertainty considered to build the scenario tree of the MS-SB NMPC does not influence the states. The scaling factor $\kappa_x^\star$ is given by $||\hat{\boldsymbol{\kappa}}_x^\star||_\infty$.

This makes sure that the box over-approximation of the scaled state ellipsoids encloses the reachable set of model (1) provided the optimization problem Equation (8) is solved to its global optimum. It was shown to be feasible in [58], where an optimization problem similar to Equation (8) was solved to its global optimum.

The formulation of the MS-SB NMPC problem that is solved at time $t$ ($s^{\text{th}}$ iteration), where $J_{so}$ represents the value of the objective function of MS-SB NMPC until the robust horizon reads as Equation (10):

$$\min_{\boldsymbol{x}_k^j, \boldsymbol{u}_k^j, \boldsymbol{d}^j, \boldsymbol{x}_{m,k}^j, \boldsymbol{X}_{c,k}^j} J_{so}(X_{k+1}^q, \Delta\boldsymbol{u}_k^j) + \sum_{k=s+N_r}^{s+N_p-1} \sum_{j=1}^{N_b^{N_r}} \omega_{k+1}^j l(\boldsymbol{x}_{k+1}^j, \Delta\boldsymbol{u}_k^j), \quad \forall (j, k+1) \in I_{st}, \tag{10a}$$

subject to:

Equations (5b)–(5f),

$$\mathcal{D} = \$(\boldsymbol{d}_0, \boldsymbol{P}_0), \tag{10b}$$

$$\left(\boldsymbol{x}_{m,k+1}^q, \boldsymbol{X}_{c,k+1}^q\right) = \mathbb{U}(\kappa_{x,k}, \boldsymbol{x}_{k+1}^{[(q-1)N_b+1:qN_b]}), \qquad \forall k \in I_{br}, q \in I_b(k), \tag{10c}$$

$$\underline{\boldsymbol{x}}_{m,k+1}^q = \boldsymbol{x}_{m,k+1}^q - \text{diag}^{\frac{1}{2}}(\boldsymbol{X}_{c,k+1}^q), \qquad \forall k \in I_{br}, q \in I_b(k), \tag{10d}$$

$$\overline{\boldsymbol{x}}_{m,k+1}^q = \boldsymbol{x}_{m,k+1}^q + \text{diag}^{\frac{1}{2}}(\boldsymbol{X}_{c,k+1}^q), \qquad \forall k \in I_{br}, q \in I_b(k), \tag{10e}$$

$$X_{k+1}^q = \boldsymbol{x}_{m,k+1}^q \cup \mathbb{C}_{vp}(\underline{\boldsymbol{x}}_{m,k+1}^q, \overline{\boldsymbol{x}}_{m,k+1}^q), \qquad \forall k \in I_{br}, q \in I_b(k), \tag{10f}$$

$$\boldsymbol{g}(\boldsymbol{x}, \boldsymbol{u}_k^{(q-1)N_b+1}) \leq 0, \qquad \forall k \in I_{br}, q \in I_b(k), \boldsymbol{x} \in X_{k+1}^q. \tag{10g}$$

where $I_b(k) := \{1, \cdots, N_b^{k-s}\}$. The scenario tree of MS-SB NMPC considers $N_b = 2n_d + 1$ branches at each node. The sigma points of the ellipsoidal uncertainty set are obtained from Equation (10b). $\boldsymbol{x}_{m,k+1}^q$ and $\boldsymbol{X}_{c,k+1}^q$ represent the state mean and the state covariance matrix obtained while applying control input $\boldsymbol{u}_k^{(q-1)N_b+1}$ at the parent state $\boldsymbol{x}_k^{p((q-1)N_b+1)}$. Referring to Figure 2, $\boldsymbol{X}_{c,2}^2$ is computed using the state predictions $\boldsymbol{x}_2^4, \boldsymbol{x}_2^5$ and $\boldsymbol{x}_2^6$ that obtained from the parent state $\boldsymbol{x}_1^2$ when the control input $\boldsymbol{u}_1^4$ ($\boldsymbol{u}_1^4 = \boldsymbol{u}_1^5 = \boldsymbol{u}_1^6$ due to non-anticipativity constraints) is applied. The state covariance matrix is scaled using a scaling factor $\kappa_{x,k}$ which is a tuning parameter. The scaling factor $\kappa_{x,k}$ must be chosen in a way that the box over-approximation of the state ellipsoid described by the state mean and the state covariance matrix encloses the reachable states sets of model (1) along the robust horizon. The scaling factor $\kappa_{x,s}$ for the first stage can be obtained by solving the optimization problem Equation (8) and is given as $\kappa_x^\star$. At the next stage, the true plant state can be present anywhere in the predicted box over-approximation of the reachable states set but the scenario tree of MS-SB NMPC branches from the predictions obtained using the sigma points. This introduces additional uncertainty in the initial condition of the states and can be overcome by increasing the scaling factor $\kappa_{x,k}$ by a constant factor $\beta$ (i.e., $\kappa_{x,k} = \beta\kappa_{x,k-1}$ s.t. $\beta \geq 1$). The lower and upper bounds on the state predictions can be obtained using Equations (10d) and (10e). The vertices of the box over-approximation of state ellipsoid along with its center are obtained in Equation (10f). The constraint function $\boldsymbol{g}(\cdot)$ is satisfied for the vertices of the box over-approximation of the predicted state ellipsoids using Equation (10g). The objective function of the MS-SB NMPC scheme is given in Equation (10a), and reads as Equation (11):

$$J_{so}(X_{k+1}^q, \Delta\boldsymbol{u}_k^j) = \sum_{k=s}^{s+N_r-1} \sum_{q=1}^{N_b^{k-s+1}} \sum_{i=1}^{2^{n_x}+1} \omega_{k+1}^{i+q-1} L(\boldsymbol{x}^i, \Delta\boldsymbol{u}_k^{(q-1)N_b+1}), \tag{11}$$

where $\boldsymbol{x}^i$ represents the $i^{\text{th}}$ element of the set $X_{k+1}^i$. $\omega_k^{i+q-1}$ is the weight associated with vertices of the box over-approximation of the state ellipsoids along with its mean and is chosen such that Equation (12):

$$\sum_{q=1}^{N_b^{k-s+1}} \sum_{i=1}^{2^{n_x}+1} \omega_k^{i+q-1} = 1, \forall k \in I_{br}. \tag{12}$$

**Remark 3.** *The tuning of scaling factor $\kappa_{x,k}$ plays a major role in the performance of MS-SB NMPC. If a very large value for $\kappa_{x,k}$ is chosen, the reachable states of model (1) are loosely over-approximated and this results in performance loss. On the contrary, if a very small value is chosen, the reachable states are under approximated and this may result in a constraint violation. The scaling factor $\kappa_{x,s}$ can be obtained by solving optimization problem 8 so that the reachable states set is tightly over-approximated by a box. The robust constraint satisfaction of MS-SB NMPC is guaranteed only if the MS-SB NMPC optimization problem Equation (10) is recursively feasible and the following condition is satisfied:*

$$\max_{\mathbf{x} \in \mathcal{X}_{k+1}^q} \mathbf{g}(\mathbf{x}, \mathbf{u}_k^{(q-1)N_b+1}) \geq \max_{\mathbf{x} \in \mathbb{X}_{k+1}^q} \mathbf{g}(\mathbf{x}, \mathbf{u}_k^{(q-1)N_b+1}), \tag{13}$$

*where $\mathbb{X}_{k+1}^q$ represents the reachable states set of model (1). The condition Equation (13) is satisfied if the predicted box over-approximation of the state ellipsoid encloses the set of reachable states and the sensitivity of the constraints with respect to the states does not change its sign. Otherwise, $\kappa_{x,0}$ should be tuned based on posterior analysis using simulation studies such that the condition Equation (13) is satisfied. In this work, the recursive feasibility of multi-stage NMPC using sigma points is verified using simulation studies. The scaling factor $\kappa_{x,k}$ is proportional to the value of the parameter $\beta$. The parameter $\beta$ is chosen based on posterior analysis using simulation studies in a way that the predicted state ellipsoids enclose the reachable states sets along the robust horizon. Theoretical guarantees on the stability and recursive feasibility of the multi-stage NMPC using sigma points are not considered in this work but can be achieved by choosing a full robust horizon (i.e., $N_r = N_p$) and tuning $\beta$ systematically [17,59]. This will be part of our future work.*

**Remark 4.** *The performance of multi-stage NMPC using sigma points can be improved using an ellipsoidal over-approximation of the reachable states set and satisfying the constraints for all states contained in the ellipsoidal set [60]. One of the major drawbacks of this approach is that if $n_x > 2n_d + 1$, the number of sigma points chosen to build the scenario tree may not be sufficient to capture the covariance matrix of the nonlinear transformation and may lead to degenerate ellipsoids. In this case, it is not feasible to compute an ellipsoidal over-approximation of the reachable states set using $2n_d + 1$ sigma points but a box over-approximation can still be computed using the scaled covariance matrix as explained in Section 3.2. An ellipsoidal over-approximation of the reachable set of states can be computed by choosing $n_x + 1$ sigma points from the uncertainty set instead of $2n_d + 1$ sigma points.*

### 3.2.2. Multi-Stage NMPC Based on the Box Over-Approximation of the Reachable Set of the Constraint Function (MS-CB NMPC)

MS-CB NMPC computes a box over-approximation of the reachable set of the constraint function $\mathbf{g}(\cdot)$ using the predictions obtained from the scenario tree of the multi-stage NMPC. The difference between MS-VA NMPC (Section 3.1.2), MS-SB NMPC (Section 3.2.1) and MS-CB NMPC (Section 3.2.2) is shown in Figure 4 for a system with two states, two uncertain parameters, a given control input and two constraints at time $k = 1$ in the prediction horizon. Figure 4a is same as the Figure 3a. Figure 4b–d represent the reachable sets in the constraint space. The pink shaded region represents the image of the constraint functions for the reachable states of the model (1). The control inputs $\mathbf{u}_1^1, \mathbf{u}_1^2, \mathbf{u}_1^3, \mathbf{u}_1^4$ and $\mathbf{u}_1^5$ are equal due to the non-anticipativity constraint.

Black dots in Figure 4b represent the value of the constraint function for the state predictions obtained on the branches of the scenario tree of MS-VA NMPC. It is assumed that $\mathbf{g}(\mathbf{x}, \mathbf{u}) \leq 0$. The optimal control input computed using MS-VA NMPC satisfies the constraints only for the parameter combinations considered in its scenario tree. The box over-approximation of the parametric

confidence region introduces additional uncertainty in the parameter values which may result in a large back-off from the bounds (shown for $g_{[2]}(x,u)$ in Figure 4b) and may result in a loss of performance.

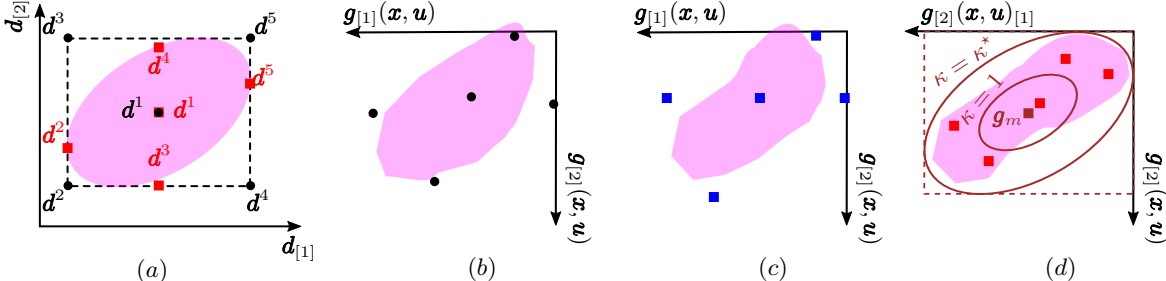

**Figure 4.** Comparison between multi-stage NMPC based on the vertex approximation, multi-stage NMPC based on the box over-approximation of the set of reachable states, multi-stage NMPC based on the box over-approximation of the reachable set of the constraint function. (**a**) Confidence region of the uncertain parameters. (**b**) Constraint space when the system is controlled using MS-VA NMPC. ● represents the constraint function values obtained for the nodes in the scenario tree. (**c**) Constraint space when the system is controlled using MS-SB NMPC. ■ represents the constraint function value obtained using the vertices of the box over-approximation of the set of reachable states along with its state mean Equation (10f). (**d**) Constraint space when the system is controlled using MS-CB NMPC.

The blue squares in Figure 4c represent the values of the constraint functions evaluated using the state mean and vertices of the box over-approximation of the set if reachable states obtained using the MS-SB NMPC approach. The optimal control input satisfies the constraints only for the state values that are obtained using the vertices of the box over-approximation of the state ellipsoids and its mean. This may lead to constraint violation and can be avoided as mentioned in Remark 3. In addition, the box over-approximation of the set of reachable states may add an additional region and may lead to an additional loss of performance (can be seen for $g_{[2]}(x,u)$ in Figure 4b).

The red squares in Figure 4d represent the values of the constraint function evaluated using the state predictions obtained on the branches of the scenario tree of MS-CB NMPC. The inner (brown) ellipsoid represents the ellipsoid computed using the unscented transformation with scaling factor $\kappa = 1$. The constraint covariance matrix is then scaled using the scaling factor $\kappa$ so that the box over-approximation (brown dotted lines) of the extended ellipsoidal set (outer brown ellipsoid) over-approximates the reachable set of the constraint function. MS-CB NMPC computes optimal control inputs such that all the elements within the reachable set of the constraint function satisfy $g(x,u) \leq 0$ as shown in Figure 4d even in the presence of a nonlinear model and nonlinear constraints, in contrast to the MS-VA and MS-SB NMPC approaches.

Tuning of the Scaling Factor $\kappa$

The scaling factor $\kappa$ can be computed in a way that the box over-approximation of the constraint function ellipsoid represented by scaled constraint covariance matrix and mean tightly over-approximates the reachable set of the constraint function $g(\cdot)$ by solving $n_c$ optimization problem similar to Equation (8), where $n_c$ represents the number of constraints. The vector $\hat{\kappa}_x$ in Equation (8a) is replaced with $\hat{\kappa}_c$ and the model predictions obtained in Equations (8b) and (8d) are propagated through the constraint function along with the control input $u_0^{wc}$ to compute the box over-approximation of the reachable set of the constraint function. The final scaling factor $\kappa_c^\star$ is then given by $||\hat{\kappa}_c^\star||_\infty$. The full formulation of the optimization problem solved to obtain the scaling factor $\kappa_c^\star$ is provided in the Supplementary Material in Section S3. The optimization problems must be solved to their global optimum [61,62] which of course is computationally costly but are computed once before solving the NMPC optimization problem.

The formulation of MS-CB NMPC optimization problem solved at time $t$ ($s^{\text{th}}$ iteration) reads as Equation (14):

$$\min_{\boldsymbol{x}_k^j, \boldsymbol{u}_k^j, \boldsymbol{d}^j, \boldsymbol{g}_{m,k}^q, \boldsymbol{G}_{c,k}^q} \sum_{k=s}^{s+N_r-1} \sum_{j=1}^{N_b^{k-s+1}} \omega_{k+1}^j l(\boldsymbol{x}_{k+1}^j, \Delta\boldsymbol{u}_k^j) + \sum_{k=s+N_r}^{s+N_p-1} \sum_{j=1}^{N_b^{N_r}} \omega_{k+1}^j l(\boldsymbol{x}_{k+1}^j, \Delta\boldsymbol{u}_k^j), \quad \forall (j, k+1) \in I_{st}, \quad (14\text{a})$$

subject to:

Equations (5b)–(5f), (10b),

$$\boldsymbol{g}_{n,k+1}^j = \boldsymbol{g}(\boldsymbol{x}_{k+1}^j, \boldsymbol{u}_k^j), \qquad\qquad\qquad \forall (j, k+1) \in I_{st}, \qquad (14\text{b})$$

$$\left(\boldsymbol{g}_{m,k+1}^q, \boldsymbol{G}_{c,k+1}^q\right) = \mathbb{U}(\kappa_{c,k}, \boldsymbol{g}_{n,k+1}^{[(q-1)N_b+1:qN_b]}), \qquad \forall k \in I_{br}, q \in I_b(k), \qquad (14\text{c})$$

$$\boldsymbol{g}_{m,k+1}^q + \text{diag}^{\frac{1}{2}}(\boldsymbol{G}_{c,k+1}^q) \leq 0, \qquad\qquad \forall k \in I_{br}, q \in I_b(k), \qquad (14\text{d})$$

The scenario tree of MS-CB NMPC considers $N_b = 2n_d + 1$ branches at each node. The value of the constraint function evaluated at each node is given by Equation (14b). The mean ($\boldsymbol{g}_m$) and the covariance of the constraint function ($\boldsymbol{G}_c$) are obtained using Equation (14c). The constraint covariance matrix is scaled using a scaling factor $\kappa_{c,k}$. The $\kappa_{c,s}$ (scaling factors for the first stage) is chosen as $\kappa_c^{\star}$ and is obtained by solving $n_c$ offline optimization problems as explained before. The scaling factor after the first stage $\kappa_{c,k}$ is chosen as $\beta\kappa_{c,k-1}$, where $\beta \geq 1$. The constraint Equation (14d) ensures that all the elements inside the box over-approximation of the constraint function reachable set satisfy the constraints. The objective function Equation (14a) of MS-CB NMPC is similar to the standard multi-stage NMPC Equation (5a). MS-CB NMPC guarantees robust constraint satisfaction if the scaling factor $\kappa_c^{\star}$ is obtained by solving an offline global optimization problem and the optimization problem Equation (14) is recursively feasible. The recursive feasibility of the MS-CB NMPC optimization problem must be verified using simulation studies.

A short comparison between the complexity of the different robust multi-stage NMPC optimization problems is given in Table 1. The optimization variables required for the computation of the model predictions, the control inputs, and the unscented transformation (mean and covariance matrix) and the constraints, and the unscented transformation and constraints for enforcing control bounds, non-anticipativity constraints, additional constraints on all nodes in the scenario tree and the reachable sets (obtained using the mean and the covariance matrix) are reported in Table 1. The scenario tree of MS NMPC and MS-VA NMPC considers $3^{n_d}$ and $2^{n_d} + 1$ branches at each node whereas MS-SB and MS-CB NMPC schemes consider $2n_d + 1$ branches at each node in their scenario tree. The number of nodes considered in the scenario tree of all formulations is given by $(N_b^{N_r} - 1)/(N_b - 1) + N_b^{N_r}(N_p - N_r + 1)$. The computational complexity of MS NMPC and MS-VA NMPC grows exponentially with respect to the number of uncertain parameters whereas it grows linearly for MS-SB and MS-CB NMPC approaches. MS-VA NMPC considers more scenarios in its scenario tree than MS-SB and MS-CB NMPC approaches for $n_d > 2$ and may require more computational effort than MS-SB and MS-CB NMPC to solve them. The MS-SB and MS-CB NMPC optimization problems require additional optimization variables for the computation of the mean and of the covariance matrix of the nonlinear transformation and are reported as OV of the unscented transformation in Table 1. The computational complexity of the MS-CB NMPC increases with the number of constraints ($n_c$) due to the computation of the constraint covariance matrix, whereas the computational complexity of MS-SB NMPC increases with the number of states ($n_x$) due to computation of the state covariance matrix. MS-SB NMPC may result in a performance loss when compared to MS-CB NMPC due to the box over-approximation of the set of reachable states. The scaling factor of MS-SB NMPC must be chosen properly for robust constraint satisfaction (see Remark 3).

**Table 1.** A complexity comparison of different robust multi-stage NMPC optimization problem. $N_{br}$ and $N_{ar}$ represent the number of nodes considered in the scenario tree for $k < N_r$ and for $N_r \le k \le N_p$ and are given by $\frac{N_b^{N_r}-1}{N_b-1}$ and $N_b^{N_r}(N_p - N_r + 1)$. $N_n = N_{br} + N_{ar}$. OV—Optimization variables.

| Number of | | MS | MS-VA | MS-SB | MS-CB |
|---|---|---|---|---|---|
| samples $(N_b)$ in set $\mathcal{D}$ | | $3^{n_d}$ | $2^{n_d}+1$ | $2n_d+1$ | $2n_d+1$ |
| **OV** | model equations control inputs unscented transformation | $N_n n_x +$ $(N_n-1)n_u$ $-$ | $N_n n_x +$ $(N_n-1)n_u$ $-$ | $N_n n_x +$ $(N_n-1)n_u$ $(n_x^2 + n_x)N_{br}$ | $N_n n_x +$ $(N_n-1)n_u$ $(n_c^2 + n_c)N_{br}$ |
| **constraints** | model equations control bounds non-anticipativity constraints additional constraint $\boldsymbol{g}(\cdot)$ unscented transformation enforcing $\boldsymbol{g}(\cdot)$ for reachable set | $N_n n_x +$ $2(N_n-1)n_u+$ $\frac{(N_b-1)^{N_r+1}-n_u}{N_b-2}$ $(N_n-1)n_c$ $-$ $-$ | $N_n n_x +$ $2(N_n-1)n_u+$ $\frac{n_u(N_b-1)^{N_r+1}-n_u}{N_b-2}$ $(N_n-1)n_c$ $-$ $-$ | $N_n n_x$ $2(N_n-1)n_u+$ $\frac{n_u(N_b-1)^{N_r+1}-n_u}{N_b-2}$ $(N_n-1)n_c$ $(n_x^2 + n_x)N_{br}$ $(2^{n_x}+1)N_{br}n_c$ | $N_n n_x +$ $2(N_n-1)n_u+$ $\frac{n_u(N_b-1)^{N_r+1}-n_u}{N_b-2}$ $(N_n-1)n_c$ $(n_c^2 + n_c)N_{br}$ $N_{br}n_c$ |

## 4. Adaptive Robust Multi-Stage NMPC

The principle of adaptive control suggests using all the information available from the system during operation to improve the performance of the controller. The measurement information can be used to improve the knowledge about the plant thereby reducing the range of the uncertainty associated with the uncertain parameters (i.e., in our case the size of the ellipsoidal set $\mathbb{D}(\boldsymbol{d}_0, \boldsymbol{P}_0)$). An estimate of the uncertain parameters can be obtained by solving a least-squares estimation problem using all the measurements collected until time $t$ (i.e., $s$ measurements from the plant). The formulation of the estimation problem reads as Equation (15):

$$\boldsymbol{d}_s = \arg\min_{\boldsymbol{d}} \sum_{k=0}^{s} (\boldsymbol{x}_{k+1}^m - \boldsymbol{x}_{k+1})^T \boldsymbol{Q} (\boldsymbol{x}_{k+1}^m - \boldsymbol{x}_{k+1}) \tag{15a}$$

subject to

$$\boldsymbol{x}_{k+1} = \boldsymbol{f}(\boldsymbol{x}_k, \boldsymbol{u}_k, \boldsymbol{d}), \qquad \forall k \in \{0, \cdots, s-1\}, \tag{15b}$$

where $\boldsymbol{d}_s$ represents the nominal value of the uncertain parameters and $\boldsymbol{Q}$ denotes the inverse of the variance-covariance matrix of the measurement noise. $\boldsymbol{x}_0$ represents the initial condition of the system and $\boldsymbol{u}_k$ denotes the sequence of control inputs that have been applied to the plant.

If we assume that the model is structurally identifiable, and white Gaussian noise is superposed on the measurements, a joint-confidence region of parameter estimates depending upon the information content of the data can be obtained according to the Cramer-Rao inequality [24]. The parametric variance-covariance matrix can be over-approximated using the inverse of the Fisher information matrix. The Fisher information matrix is given by Equation (16):

$$\boldsymbol{F}_s = \sum_{k=0}^{s} \boldsymbol{s}_k^T \boldsymbol{Q} \boldsymbol{s}_k, \tag{16}$$

where $\boldsymbol{F}_s$ denotes the Fisher information matrix obtained using measurements observed from time 0 to $t$, and $\boldsymbol{s}_k = \frac{\partial \boldsymbol{x}_k}{\partial \boldsymbol{d}}$ represents the sensitivities of the states with respect to the uncertain parameters in the presence of full state measurements. A joint-confidence ellipsoid centered at the value of the least-squares parameter estimate that bounds the values of the least-squares parameter estimates (under different realizations of the measurement noise) with a prescribed confidence level $\alpha(z) = \frac{2}{\sqrt{\pi}} \int_0^{\frac{z}{\sqrt{2}}} e^{-y^2} dy$ ( $z = 1$ corresponds to the $1\sigma$ confidence level of a Gaussian distribution) is given by Equation (17):

$$(\boldsymbol{d} - \boldsymbol{d}_s)^T \boldsymbol{F}_s (\boldsymbol{d} - \boldsymbol{d}_s) \leq n_d \, F_{dist}(n_d, s \, n_x - n_d, \alpha(z)), \tag{17}$$

where $F_{dist}$ represents the upper $\alpha(z)$ quantile of the Fisher distribution with $n_d$ and $s \, n_x - n_d$ degrees of freedom in the numerator and denominator (see Equation (18)). $\boldsymbol{P}_s$ represents the upper bound on the parameter covariance matrix with the confidence level $\alpha(z)$

$$\boldsymbol{P}_s = \frac{1}{n_d \, F_{dist}(n_d, s \, n_x - n_d, \alpha(z))} \boldsymbol{F}_s^{-1}. \tag{18}$$

We assume that the true realization of the uncertain parameter always lies within the confidence region of the uncertain parameter obtained with the chosen confidence level $\alpha(z)$ (i.e., we assume that $\mathbb{D}(\boldsymbol{d}_s, \boldsymbol{P}_s)$ has a confidence level of 100% instead of $\alpha(z)$ which is approximately satisfied if $z = 3$ sigma).

**Remark 5.** *The computational complexity of the optimization problem* (15) *increases as* $t \to \infty$. *This can be overcome by solving the parameter estimation problem* (15) *in a moving horizon fashion* [63], *where instead of using all the available measurements from the plant only the recent measurements within a window of chosen size are considered.*

### 4.1. Adaptive Standard Multi-Stage NMPC

The adaptive standard multi-stage NMPC (A-MS) and adaptive multi-stage NMPC based on the vertex over-approximation (A-MS-VA) update their scenario trees based on the observed plant measurements. A tight box over-approximation of the intersection between the initial ellipsoidal confidence region $\mathbb{D}(\boldsymbol{d}_0, \boldsymbol{P}_0)$, and the ellipsoidal confidence region can be obtained using the observed measurements, by solving the following optimization problem (Equation (19)).

$$\underline{\boldsymbol{d}}_s, \bar{\boldsymbol{d}}_s = \arg \min_{\underline{\boldsymbol{d}}_o, \bar{\boldsymbol{d}}_o \forall o \in I_d} - \sum_{o=1}^{n_d} \underline{\boldsymbol{d}}_{o,[o]} + \sum_{o=1}^{n_d} \bar{\boldsymbol{d}}_{o,[o]} \tag{19a}$$

subject to

$$(\underline{\boldsymbol{d}}_o - \boldsymbol{d}_0)^T \boldsymbol{P}_0^{-1} (\underline{\boldsymbol{d}}_o - \boldsymbol{d}_0) \leq 1, \qquad\qquad \forall o \in I_d, \tag{19b}$$

$$(\underline{\boldsymbol{d}}_o - \boldsymbol{d}_s)^T \boldsymbol{P}_s^{-1} (\underline{\boldsymbol{d}}_o - \boldsymbol{d}_s) \leq 1, \qquad\qquad \forall o \in I_d, \tag{19c}$$

$$(\bar{\boldsymbol{d}}_o - \boldsymbol{d}_0)^T \boldsymbol{P}_0^{-1} (\bar{\boldsymbol{d}}_o - \boldsymbol{d}_0) \leq 1, \qquad\qquad \forall o \in I_d, \tag{19d}$$

$$(\bar{\boldsymbol{d}}_o - \boldsymbol{d}_s)^T \boldsymbol{P}_s^{-1} (\bar{\boldsymbol{d}}_o - \boldsymbol{d}_s) \leq 1, \qquad\qquad \forall o \in I_d. \tag{19e}$$

where $I_d := \{1, \cdots, n_d\}$. $n_d$ parameter combinations ($\underline{\boldsymbol{d}}_o$ and $\bar{\boldsymbol{d}}_o$) are chosen to compute the lower and upper bounds on the ellipsoidal intersection region [64].

The formulation of the A-MS NMPC optimization problem solved at time $t$ ($s^{th}$ iteration) is similar to the MS NMPC optimization problem Equation (5), where Equation (5g) and Equation (5h) are replaced by Equation (20):

$$\underline{\boldsymbol{d}}_{[o]} = \underline{\boldsymbol{d}}_{s,[o]}, \; \bar{\boldsymbol{d}}_{[o]} = \bar{\boldsymbol{d}}_{s,[o]}, \qquad\qquad \forall o \in I_d, \tag{20a}$$

$$\mathcal{D} = \mathbb{C}_a(\underline{\boldsymbol{d}}, \boldsymbol{d}_s, \bar{\boldsymbol{d}}). \tag{20b}$$

The formulation of the A-MS-VA NMPC optimization problem solved at time $t$ ($s^{th}$ iteration) is similar to the A-MS NMPC problem, where instead of Equation (20) the following equations (Equation (21)) are used

$$\underline{\boldsymbol{d}}_{[o]} = \underline{\boldsymbol{d}}_{s,[o]}, \; \bar{\boldsymbol{d}}_{[o]} = \bar{\boldsymbol{d}}_{s,[o]}, \qquad\qquad \forall o \in I_d, \tag{21a}$$

$$\mathcal{D} = \boldsymbol{d}_s \cup \mathbb{C}_{vp}(\underline{\boldsymbol{d}}, \bar{\boldsymbol{d}}). \tag{21b}$$

$\underline{d}$ and $\bar{d}$ represent the lower and upper bounds on the uncertain parameters which bound the intersection of the ellipsoidal confidence regions $\mathbb{D}(d_0, P_0)$ and $\mathbb{D}(d_s, P_s)$. Whenever a measurement from the plant becomes available, the new parameter estimates are obtained by solving the least-squares estimation problem Equation (15), and the corresponding parameter covariance matrix is computed using Equation (18). The optimization problem Equation (19) is then solved to compute the new bounds on the uncertain parameters. The scenario trees of the A-MS/A-MS-VA NMPC optimization problem are updated according to Equation (20) or Equation (21) and the adapted optimization problem is solved to compute the optimal control input which is applied to the plant.

## 4.2. Adaptive Multi-Stage NMPC Using Sigma Points

The adaptive multi-stage NMPC formulation based on the box over-approximation of the reachable states set (A-MS-SB) and the adaptive multi-stage NMPC formulation based on the box over-approximation of the reachable constraint function set (A-MS-CB) also use updated information about the uncertain parameters to improve the performance of multi-stage NMPC using sigma points. The true value of the uncertain parameters will be contained in the intersection region between the ellipsoidal confidence region used in the previous A-MS-SB and A-MS-CB NMPC optimization problem, and the confidence region of the uncertain parameters obtained in addition using the current plant measurements ($\mathbb{D}(d_s, P_s)$). We can compute an ellipsoidal over-approximation of the intersection region between two ellipsoids as shown in [65] and then update the scenario tree of A-MS-SB and A-MS-CB NMPC using the sigma points that result from the over-approximating ellipsoid.

The formulations of A-MS-SB NMPC and A-MS-CB NMPC are similar to MS-SB NMPC and MS-CB NMPC Equations (10) and (14) where Equation (10b) are replaced by the following Equation (22):

$$P_s^\star = (1 - \phi(1 - \phi)(d_s - d_{s-1}^\star)^T P_s^{-1} \hat{F}^{-1}(P_{s-1}^\star)^{-1}(d_s - d_{s-1}^\star))\hat{F}^{-1}, \tag{22a}$$

$$d_s^\star = \hat{F}^{-1}(\phi(P_{s-1}^\star)^{-1} d_{s-1}^\star + (1 - \phi)P_s^{-1} d_s), \tag{22b}$$

$$\mathcal{D} = \mathbb{S}(d_s^\star, P_s^\star), \tag{22c}$$

where $\hat{F} = \phi(P_{s-1}^\star)^{-1} + (1 - \phi)P_s^{-1}$. $d_s^\star$ and $P_s^\star$ give the mean and the covariance matrix of an over-approximating ellipsoid which over-approximates the intersection between $\mathbb{D}(d_{s-1}^\star, d_{s-1}^\star)$ and $\mathbb{D}(d_s, d_s)$, and are obtained using Equations (22a) and (22b). At time $t = 0$, $d_{s-1}^\star = d_0$ and $P_{s-1}^\star = P_0$. $\phi$ is a degree of freedom of the A-MS-SB and A-MS-CB NMPC optimization problem and is bounded between 0 and 1. The scenario tree is generated using the parameter combinations given as the sigma points of the over-approximating ellipsoid $\mathbb{D}(d^\star, P^\star)$, see Equation (22c).

**Remark 6.** *The key difference between the adaptive version of multi-stage NMPC (A-MS NMPC and A-MS-VA NMPC) and multi-stage NMPC using sigma points (MS-SB NMPC and MS-CB NMPC) lies in the approximation of the intersection region between two ellipsoids to generate the scenario tree. Standard adaptive multi-stage NMPC approaches over-approximate the intersection region by a minimum perimeter box which can be obtained by solving the optimization problem Equation (19). The adaptive version of multi-stage NMPC using sigma points over-approximates the intersection region by an ellipsoid, which is computed as a part of the NMPC optimization problem. In the NMPC optimization problem an over-approximating ellipsoid from a set of candidate ellipsoids is computed that is optimal for adaptive multi-stage NMPC using the sigma points approaches. This comes at the expense of an additional computational cost.*

**Remark 7.** *The tuning parameters $\kappa_s$, $\kappa_c$, and $\kappa_o$ that are obtained by solving an optimization problem a-priori as mentioned in Sections 3.2.1 and 3.2.2 are valid only for the robust multi-stage NMPC using sigma points. In the A-MS-SB and A-MS-CB optimization problems, ellipsoidal over-approximations of the intersection region are computed as part of the robust NMPC optimization problem. Here, the scaling factors $\kappa_s$, $\kappa_c$, and $\kappa_o$ have to be tuned based on trial-error using simulation studies.*

## 5. Case Studies

The performance of the different robust multi-stage NMPC controllers is compared using two case studies. A simple benchmark semi-batch reactor example adapted from [66] is chosen to perform a detailed analysis of the performance of robust multi-stage NMPC for different values of the tuning parameters. A more complex industrial semi-batch polymerization reactor [67] is chosen to demonstrate the performance of different control schemes. The controllers described in Section 3 are referred to as non-adaptive controllers and the controllers described in Section 4 as adaptive controllers.

### 5.1. Case Study 1: Benchmark Semi-Batch Reactor

The semi-batch reactor benchmark example from [66] is adapted to investigate the advantages of the proposed schemes. The chemical reaction under consideration is given by

$$A + B \rightarrow C.$$

The nonlinear model is obtained from the mass balance of the reactor, molar balances of the reactants A and B, and energy balances of the reactor and the jacket. The model equations are given below Equation (23):

$$\dot{V}_R = \dot{V}_{in}, \tag{23a}$$

$$\dot{c}_A = -\frac{\dot{V}_{in}}{V_R} c_A - K c_A c_B, \tag{23b}$$

$$\dot{c}_B = \frac{\dot{V}_{in}}{V_R} (c_{B,in} - c_B) - K c_A c_B, \tag{23c}$$

$$\dot{T}_R = \frac{\dot{V}_{in}}{V_R} (T_{in} - T) - \frac{\alpha A_w (T_R - T_J)}{\rho V_R c_p} - \frac{K c_A c_B H}{\rho c_p}, \tag{23d}$$

$$\dot{T}_J = \frac{\dot{Q}_K + \alpha A_w (T_R - T_J)}{\rho V_J c_p}, \tag{23e}$$

with

$$c_C = \frac{c_{A,0} V_{R,0} + c_{C,0} V_{R,0} - c_A V_R}{V_R}, \tag{23f}$$

$$A_w = \pi r^2 + \frac{0.002 V_R}{r}, \tag{23g}$$

where $V_R$ denotes the volume of the reactor, $c_A$, $c_B$, and $c_C$ represent the concentrations of the reactants A and B and the product C. $T_R$, and $T_J$ represent the temperature of the contents inside the reactor and the jacket. $A_w$ denotes the inner surface area of the reactor covered with the reaction mixture. The feed ($\dot{V}_{in}$) and the cooling power of the jacket ($\dot{Q}_k$) are the control inputs.

### 5.1.1. Formulation of the Optimal Control Problem

The control task is to maximize the number of moles of product C that are produced along the prediction horizon while satisfying the constraints until the specified end time of 1.0 h is reached. The sampling time $t_s$ is chosen as 0.05 h. The temperature of the reactor $T_R$ must be kept between 322 K and 326 K. The volume of the reactor contents must not exceed 7 L. It is assumed the specific reaction rate constant $K$ and the reaction enthalpy $H$ are not known precisely. The parameters are contained in an ellipsoidal confidence region described by their nominal value and the parameter covariance matrix (see Equation (24)).

$$d_0 = \begin{pmatrix} H \\ K \end{pmatrix} = \begin{pmatrix} -355 \frac{kJ}{mol} \\ 1.205 \frac{l}{mol\,h} \end{pmatrix}, P_0 = \begin{pmatrix} 11300 & -7.7 \\ -7.7 & 0.131 \end{pmatrix}. \tag{24}$$

All other parameters in the model equations are considered to be known and are given along with a short description in Table 2. All the states ($\boldsymbol{x} = (V_R, c_A, c_B, T_R, T_J)^T$) are assumed to be measured with measurement noise of standard deviation $\boldsymbol{\sigma} = (0.0001\ \text{L},\ 0.01\ \text{mol L}^{-1},\ 0.01\ \text{mol L}^{-1},\ 0.1\ \text{K},\ 0.1\ \text{K})^T$. The control vector ($\boldsymbol{u}$) is represented as $(\dot{V}_{in}, \dot{Q}_k)^T$. The bounds on the control inputs are given in Table 3. The states are not constrained but reasonable bounds on the states are chosen to reduce the computational effort. The bounds on the states along with their initial values are given in Table 4.

**Table 2.** Case study 1: Model parameters.

| Parameter | Description | Value | Unit |
|---|---|---|---|
| $\alpha$ | heat-transfer coefficient between the reactor and jacket | 1700 | $\text{kJ K}^{-1}\,\text{h}^{-1}\,\text{m}^{-2}$ |
| $r$ | radius of the cross-section of the inner part | 0.092 | m |
| $\rho$ | density of the reactor contents | 1000 | $\text{g L}^{-1}$ |
| $c_p$ | specific heat capacity of the reactor contents | 4.2 | $\text{J g}^{-1}\,\text{K}^{-1}$ |
| $c_{B,in}$ | input concentration of reactant B | 3 | $\text{mol L}^{-1}$ |
| $V_J$ | volume of the contents inside the cooling jacket | 2.22 | L |
| $T_{in}$ | temperature of the flows entering the reactor | 300 | K |
| $c_{C,0}$ | initial concentration of the product C | 0 | $\text{mol L}^{-1}$ |

**Table 3.** Case study 1: Bounds on the control inputs.

| Control | Lower Bound | Upper Bound | Unit |
|---|---|---|---|
| $\dot{V}_{in}$ | 0 | 32.4 | $\text{L h}^{-1}$ |
| $\dot{Q}_K$ | $-9000$ | 0 | $\text{kJ h}^{-1}$ |

**Table 4.** Case study 1: Initial conditions of the states along with practical bounds.

| State | Initial Condition | Lower Bound | Upper Bound | Unit |
|---|---|---|---|---|
| $V_R$ | 3.5 | 0 | 8 | L |
| $c_A$ | 2 | 0 | 5 | $\text{mol L}^{-1}$ |
| $c_B$ | 0 | 0 | 5 | $\text{mol L}^{-1}$ |
| $T_R$ | 325 | 273 | 350 | K |
| $T_J$ | 325 | 273 | 350 | K |

The nominal MPC optimization problem that is solved at time $t$ ($s^{\text{th}}$ iteration) reads as Equation (25):

$$\min_{\boldsymbol{x}_k, \boldsymbol{u}_k, \boldsymbol{\epsilon}_k} \sum_{k=s}^{s+N_p-1} -c_C V_R + 0.0154(\Delta \dot{V}_{in,k})^2 + 5.5 \times 10^{-5}(\Delta \dot{Q}_k)^2 + 10^6 \epsilon_{k,[1]}^2 + 10^{10}\epsilon_{k,[2]}^2, \tag{25a}$$

subject to

$$\boldsymbol{x}_{k+1} = \boldsymbol{f}(\boldsymbol{x}_k, \boldsymbol{u}_k, \boldsymbol{d}_k), \tag{25b}$$

$$322 \le T_{R,k} + \epsilon_{k,[1]} \le 326, \tag{25c}$$

$$V_{R,k} + \epsilon_{k,[2]} \le 7, \tag{25d}$$

$$-1 \le \epsilon_{k,[1]} \le 1, \tag{25e}$$

$$-0.01 \le \epsilon_{k,[2]} \le 0.01, \tag{25f}$$

$$\underline{\boldsymbol{u}} \le \boldsymbol{u}_k \le \bar{\boldsymbol{u}}, \tag{25g}$$

$$\boldsymbol{x}_s = \boldsymbol{x}_s^m, \tag{25h}$$

where $\Delta \dot{V}_{in,k} = \dot{V}_{in,k-} - \dot{V}_{in,k-1}$, and $\Delta \dot{Q}_k = \dot{Q}_k - \dot{Q}_{k-1}$. The stage cost is chosen to maximize the number of moles of product C produced along the prediction horizon and to penalize the control moves and the constraint violations. The weights for penalizing the control moves are chosen to prevent

drastic changes of the inputs. The model Equation (23) is discretized using orthogonal collocation on the finite elements [68,69] and is given in Equation (25b). The collocation points are obtained using the Legendre method [70]. The  number of finite elements and the degree of the polynomial are chosen as 3 and 1. The additional constraints that must be satisfied are given by Equations (25c) and (25d). These constraints are implemented as soft constraints to prevent the MPC optimization from becoming infeasible due to the influence of measurement noise. The maximum constraint violation that can happen is bounded by Equations (25e) and (25f). The control bounds are enforced using Equation (25g), where $\underline{\boldsymbol{u}}_k$ and $\bar{\boldsymbol{u}}_k$ represent the lower and upper bounds that can be obtained from Table 3. The plant is initialized at the current plant measurements using Equation (25h), where $\boldsymbol{x}_s^m \in \mathbb{R}^5$ represents the measurement vector obtained at time $t$.

### 5.1.2. Results Obtained Using the Non-Adaptive and the Adaptive Controllers for One Set of Parameters

The reference used for the comparison of the different multi-stage NMPC strategies considers the nominal value of the uncertain parameter to be the true realization of the uncertain parameter. The prediction horizon ($N_p$) and the robust horizon ($N_r$) of the variants of multi-stage NMPC are chosen as 5 and 2. All the nodes in the scenario trees of multi-stage NMPC are equally weighted (i.e., $w_k^i = \frac{1}{N_n}$, s.t. $\sum_{i=1}^{N_n} w_k^i = 1$, where $N_n$ is the number of nodes at time step $k$ in the prediction horizon). The scaling factors $\kappa_s$ and $\kappa_c$ of the multi-stage NMPC using sigma points were obtained as mentioned in Section 3.2. The optimization problems are solved using the global optimization toolbox from MATLAB [62]. The optimal values of the tuning parameters $\kappa_s^\star$ and $\kappa_c^\star$ resulted as 1.57 and 1.56. The scaling increase factor ($\beta$) is chosen as 1.02. The value of $\alpha$ is chosen according to the $3\sigma$ range of the variables for the adaptive NMPC approaches. The plant measurements are obtained by adding white Gaussian noise with standard deviation $\sigma$ to the true plant states. The weights on the sigma points ($\boldsymbol{v}$) chosen to build the covariance matrix of the MS-SB and MS-CB NMPC are given as $(0.2, 0.2, 0.2, 0.2, 0.2)^T$.

We use the number of moles of the product C after 0.3 h of the batch time as an indicator of the performance because this is a measure of the efficiency of the process in the first phase. When the full amount of B has been fed, the further evolution of the process is determined by the kinetics alone and the final batch time depends on the specified conversion. The results obtained when the plant is controlled using the different NMPC strategies are shown in Figure 5. The green plots show the simulation results obtained when the plant is controlled using nonlinear model predictive control under the assumption that the true plant model is known. The optimal operation is to feed as much reactant B as possible while respecting the constraints Equations (25c) and (25d). The temperature of reactant B entering into the reactor ($T_{\text{in}}$) is less than the reactor temperature $T_R$, hence the reference NMPC at the beginning tries to feed as much material as possible while satisfying the lower bound on the reactor temperature. Once the reactor is filled, the reactor temperature increases due to the exothermic reaction taking place inside the reactor and the reactor temperature hits the upper bound. The reference NMPC solution maintains the reactor temperature within its limits until the batch end by manipulating its cooling power ($\dot{Q}_K$) as the true plant and the nominal plant model are identical.

MS NMPC feeds a smaller amount of reactant B due to the presence of large uncertainty in the parameters and the tight specification on the admissible reactor temperature. The scenarios generated using the upper bound on the reaction enthalpy and the lower bound on the reaction rate hit the lower constraint on the reactor temperature, whereas the scenarios generated using the lower bound on the reaction enthalpy and the upper bound on the reaction rate hit the upper constraint on the reactor temperature in the predictions and are shown in Figure 6, where the blue lines indicate the states and the control inputs predicted by the scenario tree of the MS NMPC optimization problem solved at time $t = 0.15$ h. The lower bound constraint on the reactor temperature prevents MS NMPC from using the full cooling power of the plant to increase the feed. The number of moles of reactant B fed into the reactor using MS-VA NMPC is similar to the number of moles of reactant B fed using the MS NMPC

approach because the scenario trees of both formulations include the scenarios built using the upper bound on the reaction enthalpy and the lower bound on the reaction rate as well as using the lower bound on the reaction enthalpy and the upper bound on the reaction rate which prevent the controllers from feeding more reactants into the reactor.

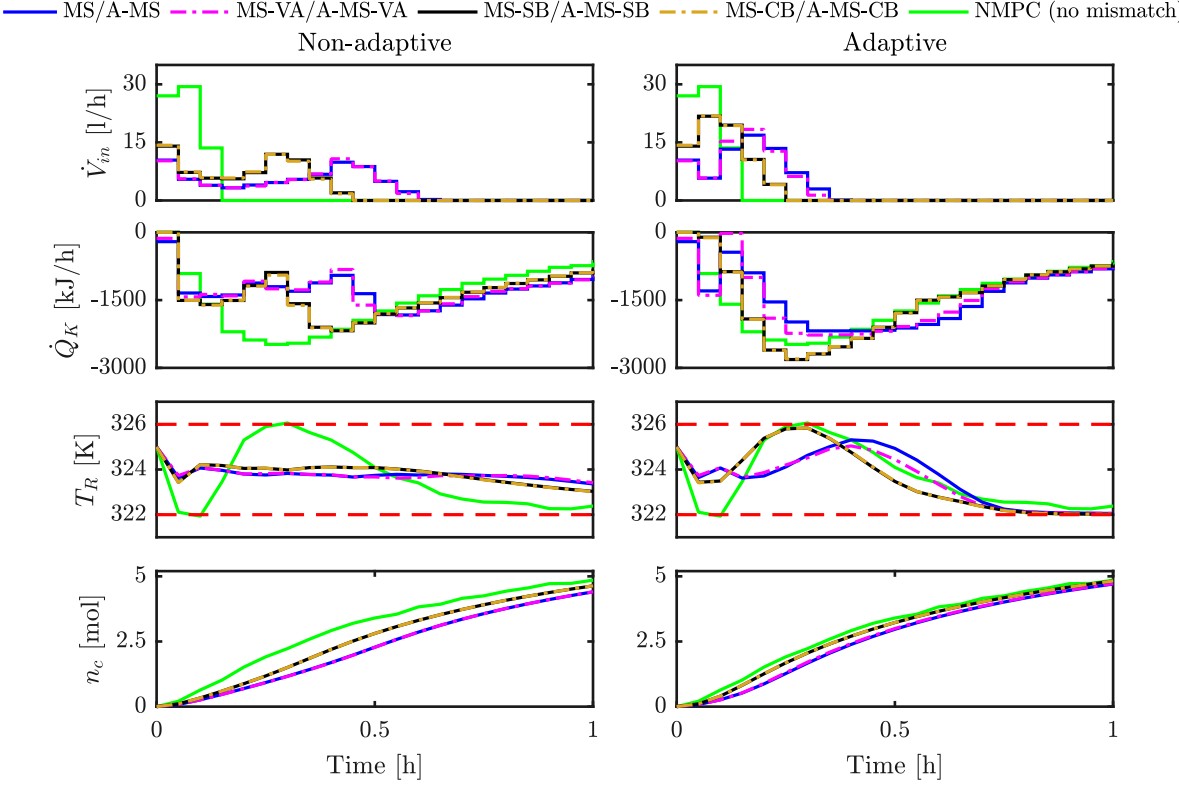

**Figure 5.** Input feed, jacket cooling power, reactor temperature, and moles of product C obtained using different NMPC strategies in the presence of measurement noise. Left figure: The plant is controlled using the non-adaptive approaches explained in Section 3, Right figure: The plant is controlled using the adaptive approaches explained in Section 4. The true plant parameters are assumed to be the nominal parameters. NMPC (no mismatch)—the plant model and the simulation model are identical.

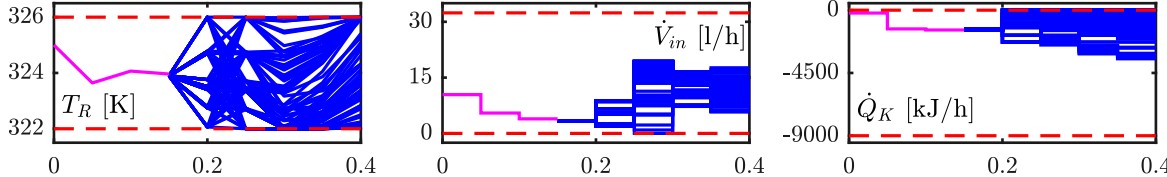

**Figure 6.** Reactor temperature, input feed, and jacket cooling power predicted in the scenario tree of the MS NMPC optimization problem solved at time $t = 0.15$ h. The magenta line (—) indicates the measured plant state and control inputs applied to the plant in the past, the blue lines (—) indicate the predicted plant states and control inputs and the dashed red lines (– –) indicate the constraints.

The number of moles of reactant B fed into the reactor is increased when using the multi-stage NMPC variants based on sigma points. There is a 29% increase in the number of moles of product C produced for the MS-SB and MS-CB NMPC variants over MS and MS-VA NMPC. The MS-SB and MS-CB NMPC variants build the scenario tree using the sigma points and tightly approximate the uncertainty set in contrast to MS and MS-VA NMPC where the scenario tree is built based on the box over-approximation of the uncertainty set. The constraints Equations (25d) and (25c) are linear with respect to the states and concern only one state ($T_R$ or $V_R$), hence the performance of the MS-SB

NMPC is similar to the MS-CB NMPC. The tuning parameter $\kappa_s$ was obtained a-priori by solving the optimization problem Equation (8) and this guarantees robust constraint satisfaction of MS-SB NMPC since the constraints are linear in the states. All the non-adaptive robust NMPC schemes satisfy the constraints on the reactor temperature and reactor volume until the batch end.

The time taken to solve one iteration of MS NMPC, MS-VA, MS-SB, and MS-CB NMPC are 1.91 s, 0.56 s, 0.63 s, and 0.44 s, in average. MS NMPC considers 9 branches at each node in the scenario tree in contrast to all the other NMPC approaches where only 5 scenarios are considered. There is a small increase in the computational time of MS-SB NMPC when compared to MS-VA NMPC due to the computation of the state covariance matrix along the robust horizon. The computation time of MS-CB NMPC is less than that of MS-SB NMPC because only two constraints $n_c = 2$ are present and the computation of the constraint covariance matrices reduces the computational effort compared to the computation of the state covariance matrix with $n_x = 5$.

The number of moles of reactants fed into the reactor in the initial phase can be improved using the adaptive NMPC approaches. There is a 47% increase in the number of moles of product C produced within the first 0.3 h when using the A-MS and A-MS-VA NMPC strategies over the MS and MS-VA NMPC approach and a 17% increase in the number of moles of product C produced from using the A-MS-SB and A-MS-CB NMPC formulations over the A-MS and A-MS-VA NMPC.

The confidence regions of the uncertain parameters at time $t = 0.05$ h ($2^{nd}$ NMPC iteration) is shown in Figure 7. The confidence region considered by A-MS-SB NMPC is similar to that of A-MS-CB NMPC, hence it is not plotted. A-MS and A-MS-VA NMPC schemes approximate the intersection between the initial confidence region and the confidence region obtained using the plant measurements by a box whereas the A-MS-SB and A-MS-CB NMPC schemes approximate the intersection region by an ellipsoid. The parameter combinations that are used to build the scenario trees of the multi-stage NMPC formulations are shown by green squares (■). The box over-approximation includes additional uncertainty by the upper and lower bounds on the uncertain parameters which result in a performance loss. The tuning parameters $\kappa_{s,0}$, and $\kappa_{c,0}$ that are computed by the non-adaptive NMPC approaches by solving an optimization problem are assumed to be valid for the adaptive approaches as well. The scaling factors are validated using the plant measurements, where it is verified whether the box over-approximation of the reachable states and the constraint function set predicted at stage $k + 1$ of A-MS-SB and A-MS-CB NMPC optimization problems solved at $(s - 1)^{th}$ iteration contains the plant measurement ($x_s^m$) observed at time $t$ and the constraint function value obtained using the plant measurements and the applied control input, respectively.

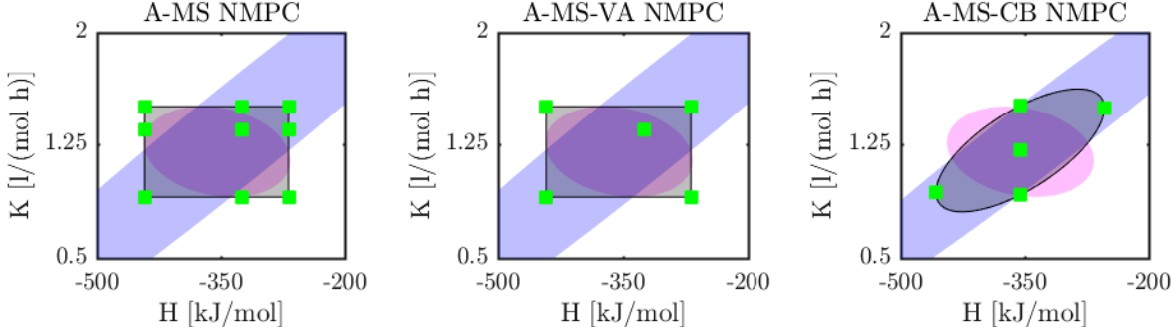

**Figure 7.** Confidence regions of the uncertain parameters considered in the adaptive NMPC variants at $t = 0.05$ h. ■ represents the initial confidence region, ■ represents the confidence region obtained using the observed measurements, ■ represents the confidence region that is considered by the adaptive NMPC schemes and ■ represents the parameter combinations that are used to build the scenario tree of the adaptive NMPC schemes.

The times taken to solve one iteration of A-MS NMPC, A-MS-VA, A-MS-SB, and A-MS-CB NMPC on average are 1.85 s, 0.57 s, 3.4 s, and 1 s. The computation time of A-MS and A-MS-VA NMPC is

similar to that of MS and MS-VA NMPC because the box over-approximation of the intersection region is computed by solving the optimization problem Equation (21) separately and then the scenario tree of the multi-stage NMPC is updated. The increase in the computational time of A-MS-SB and A-MS-CB NMPC when compared to MS-SB and MS-CB NMPC results from the computation of an ellipsoidal over-approximation of the intersection region between two ellipsoids as a part of the NMPC optimization problem.

### 5.1.3. Controller Performance Analysis for Different Values of the Tuning Parameters

The results obtained for different choices of the tuning parameters and 100 realizations of the uncertain parameters are presented in this section. The realizations are marked by black dots (•) in Figure 8. All tuning parameters other than the tuning parameter for which the controller performance is evaluated are the same as in the reference case. We evaluate the performance of the controllers by the amount of product obtained at 0.3 h. The measurements of the plant states were assumed to reduce the computational effort. The results are plotted using a violin plot [71]. The widths of the violins correspond to the normalized histogram plot. The blue color (■) represents multi-stage NNPC (MS NMPC, A-MS NMPC) using the approach in [9], the magenta color (■) represents multi-stage NMPC based on the vertex approximation (MS-VA NMPC, A-MS-VA NMPC), the black color (■) represents multi-stage NMPC based on the box over-approximation of the reachable states set (MS-SB NMPC, A-MS-SB NMPC) and the golden color (■) represents the proposed multi-stage NMPC based on the box over-approximation of the reachable set of the constraint function (MS-CB NMPC, A-MS-CB NMPC). The green diamond (◆) and the red line (—) represent the mean and the median value of the number of moles of product C that are produced at 0.3 h of batch time (after 6 NMPC iterations) using the different robust NMPC strategies.

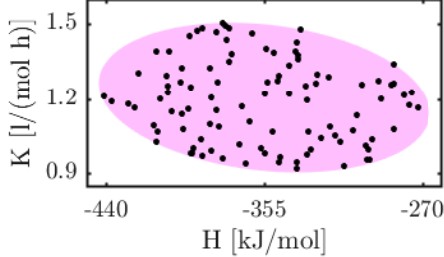

**Figure 8.** Parameter combinations considered to be the true realization of the uncertain parameters chosen within the initial confidence region of the uncertain parameters to evaluate the performance NMPC controllers for different values of the tuning parameters.

### Effect of the Length of the Prediction Horizon

The amount of product C produced during the first 0.3 h using the non-adaptive and adaptive NMPC schemes for different choices of the prediction horizon length ($N_p$) is shown using a violin plot in Figure 9. The performance of the non-adaptive and of the adaptive controllers increases significantly from $N_p = 2$ to $N_p = 3$ and then slightly up to $N_p = 5$. The NMPC formulations with shorter prediction horizon (i.e., $N_p \leq 3$) were not able to predict the performance gain achieved at the end of the batch correctly and result in a loss of performance. The adaptive NMPC schemes using the sigma points fill the reactor at 0.25 h as shown in Figure 5, hence a prediction horizon of length 5 is sufficient for them to approximate the maximization of the number of moles of product C produced at 0.3 h. The average computation time needed for solving one NMPC iteration for different lengths of the prediction horizon is indicated in Figure 10. The computational times of the adaptive and non-adaptive schemes increase with the length of the prediction horizon length.

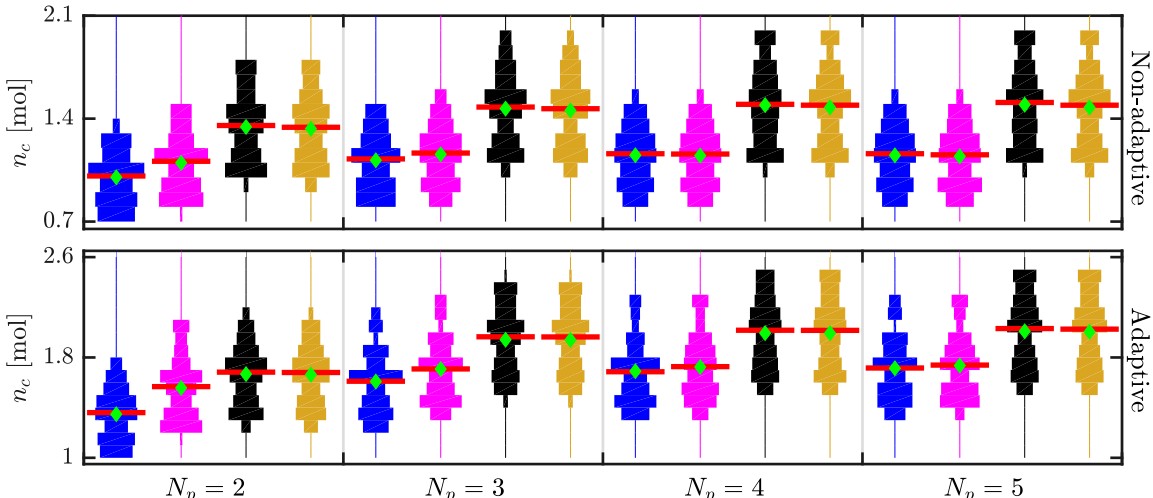

**Figure 9.** Violin plots of the amount of product C produced in the first 0.3 h in moles using non-adaptive and adaptive NMPC schemes with different lengths of prediction horizon ($N_p$) in the absence of measurement noise. ■ represents MS NMPC, A-MS NMPC, ■ represents MS-VA NMPC, A-MS-VA NMPC, ■ represents MS-SB NMPC, A-MS-SB NMPC, and ■ represents MS-CB NMPC, A-MS-CB NMPC. ◆ and ▬ represent the mean and median values of the number of moles of product C produced.

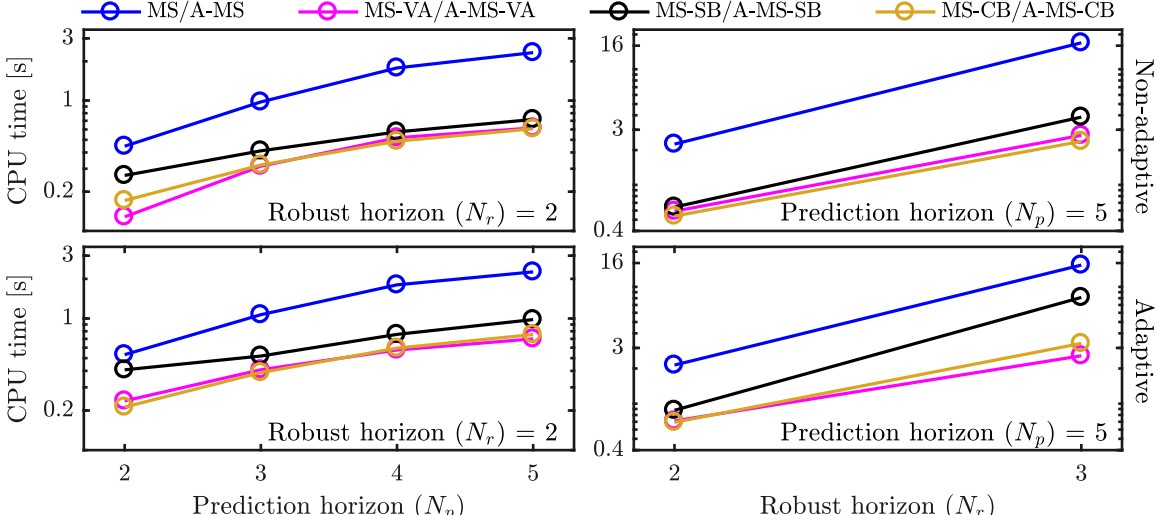

**Figure 10.** Semi-log plots of the average computation time taken per iteration [in [s]] by the different NMPC schemes for different values of the prediction horizon and robust horizon.

Effect of the Length of the Robust Horizon

The amount of product C produced during the first 0.3 h using the non-adaptive and adaptive NMPC schemes for different choices of the robust horizon length ($N_r$) is shown in Table 5. The number of moles of product C produced decreases with an increase in the length of the robust horizon. The multi-stage NMPC with $N_r = 1$ assumes that the uncertainty realization remains constant after the first stage and provides an optimistic control strategy. The non-adaptive schemes with $N_r = 1$ solved at $s = 0$ feed more reactant B into the reactor under assumptions that the uncertainty realization becomes known and remains constant after the first stage. The multi-stage NMPC solved at the next time steps $s > 1$ cannot find an optimal control input that satisfies the constraints for all scenarios along the robust horizon because of the control decision taken in the previous time step, the tight bounds on the reactor temperature and large initial confidence region of the uncertain parameters.

**Table 5.** Case study 1: Minimum (min), mean and maximum (max) number of moles of product C produced in the first 0.3 h using the non-adaptive and adaptive NMPC schemes with different lengths of robust horizon $(N_r)$ in the absence of measurement noise. A prediction horizon $(N_p)$ of length 5 is chosen.

| | Robust Horizon | Non-Adaptive | | | | Adaptive | | | |
|---|---|---|---|---|---|---|---|---|---|
| | | MS | MS-VA | MS-SB | MS-CB | A-MS | A-MS-VA | A-MS-SB | A-MS-CB |
| min | 2 | 0.83 | 0.82 | 1.07 | 1.06 | 1.37 | 1.38 | 1.52 | 1.52 |
| | 3 | 0.74 | 0.76 | 0.98 | 0.97 | 1.13 | 1.21 | 1.34 | 1.34 |
| mean | 2 | 1.15 | 1.14 | 1.49 | 1.47 | 1.71 | 1.74 | 2.00 | 2.00 |
| | 3 | 1.02 | 1.07 | 1.37 | 1.36 | 1.50 | 1.60 | 1.82 | 1.82 |
| max | 2 | 1.52 | 1.52 | 1.99 | 1.97 | 2.26 | 2.27 | 2.50 | 2.5 |
| | 3 | 1.35 | 1.41 | 1.83 | 1.81 | 2.00 | 2.15 | 2.36 | 2.35 |

For multi-stage NMPC with $N_r = 2$ the NMPC optimization problem for $s > 0$ has an optimal control input that satisfies the constraints on the reactor temperature for all the scenarios considered in its scenario tree. Multi-stage NMPC with $N_r = 3$ results in a small performance loss. A disadvantage of choosing a larger robust horizon is the exponential increase in the computational time of the multi-stage NMPC optimization problem. The computational times taken by the non-adaptive and adaptive NMPC schemes for different choices of the robust horizon are also reported in Figure 10. MS and A-MS NMPC with $N_r = 2$ consider 81 scenarios and MS-VA, A-MS-VA, MS-SB, A-MS-SB, MS-CB and A-MS-CB NMPC with $N_r = 2$ consider 25 scenarios. MS and A-MS NMPC for $N_r = 3$ consider 729 scenarios and The MS-VA, A-MS-VA, MS-SB, A-MS-SB, MS-CB and A-MS-CB NMPC with $N_r = 3$ consider 125 scenarios.

### Effect of the Scaling Factor $\kappa_k$

The covariance matrix of the multi-stage NMPC using sigma points obtained using the unscented transformation (MS-SB, A-MS-SB, MS-CB, and A-MS-CB NMPC) is scaled using the scaling factor $\kappa_k$ such that the box over-approximation of the ellipsoids represented by the covariance matrix and mean encloses the set of reachable states and constraint function values. The scaling factor consists of two parts $\kappa_0$ and $\beta$. The scaling factor $\kappa_0$ is tuned such that reachable sets at the first stage are enclosed in the prediction horizon. The scaling increase factor $\beta$ reflects the uncertainty in the initial condition of the future states after the first stage.

### Effect of Change in the Scaling Factor $\kappa_0$

The scaling factor $\kappa_0$ can be obtained using a trial-error method based on posterior analysis on simulation studies or by solving an optimization problem so that it is valid for the entire operating region of the plant. The number of moles of product C produced up to 0.3 h obtained using the MS-SB, A-MS-SB, MS-CB, and A-MS-CB NMPC formulations for different values of $\kappa_0$ are shown in Figure 11, where $\kappa_0 = \kappa^\star$ corresponds to the value of the scaling factor that was obtained by solving the optimization problem Equation (8) (i.e., $\kappa_{s,0} = 1.57$, and $\kappa_{c,0} = 1.56$).

As can be expected, the amount of product C produced in the initial phase of the batch run decreases with an increase in the value of scaling factor $\kappa_0$. The scaling factor $\kappa_0 = 1.25$ is less than $\kappa^\star$, this implies that the box over-approximation predicted at the first stage $(k = s + 1)$ is not valid for the entire operating region of the plant but it is valid for the relevant operating region since the scaling factor $\kappa_0 = 1.25$ is not invalidated by the observed measurements. It can be seen that larger values of $\kappa_0$ reduce the performance.

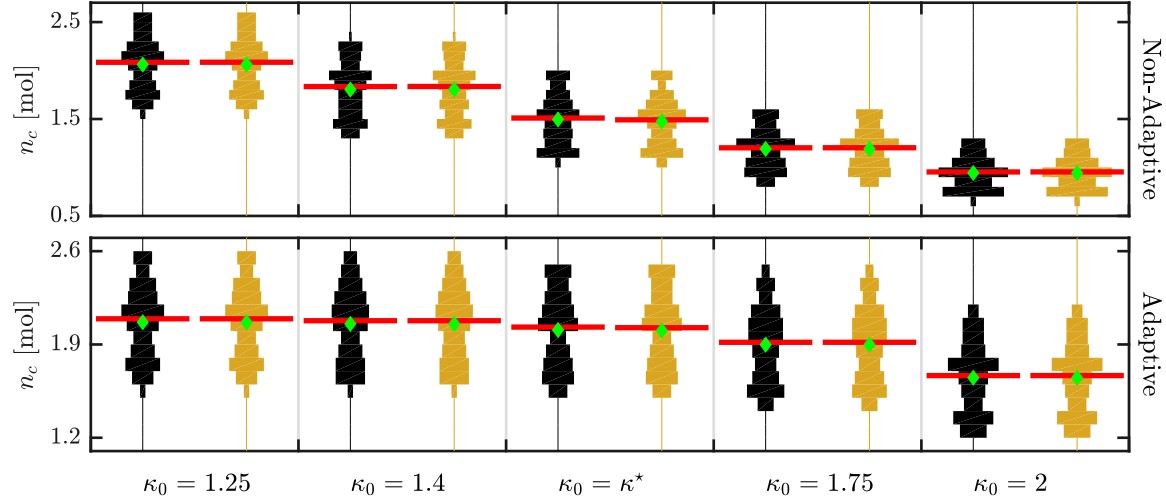

**Figure 11.** Violin plot of the amount of product C produced in the first 0.3 h using MS-SB, A-MS-SB, MS-CB, and A-MS-CB NMPC schemes with different values of the scaling factor considered at the first stage $(\kappa_0)$ in the absence of measurement noise. ■ represents MS-SB NMPC, A-MS-SB NMPC, and ■ represents MS-CB NMPC, A-MS-CB NMPC. ◆ and ▬ represent the mean and the median value.

Effect of change in scaling increase factor $\beta$

Figure 12 shows the number of moles of product C produced during the initial part of the batch run using non-adaptive and adaptive NMPC using sigma points for different values of the scaling increase factor $\beta$. The amount of product C decreases significantly with an increase in the value of $\beta$ for the non-adaptive versions, while for the adaptive cases, the influence is minor.

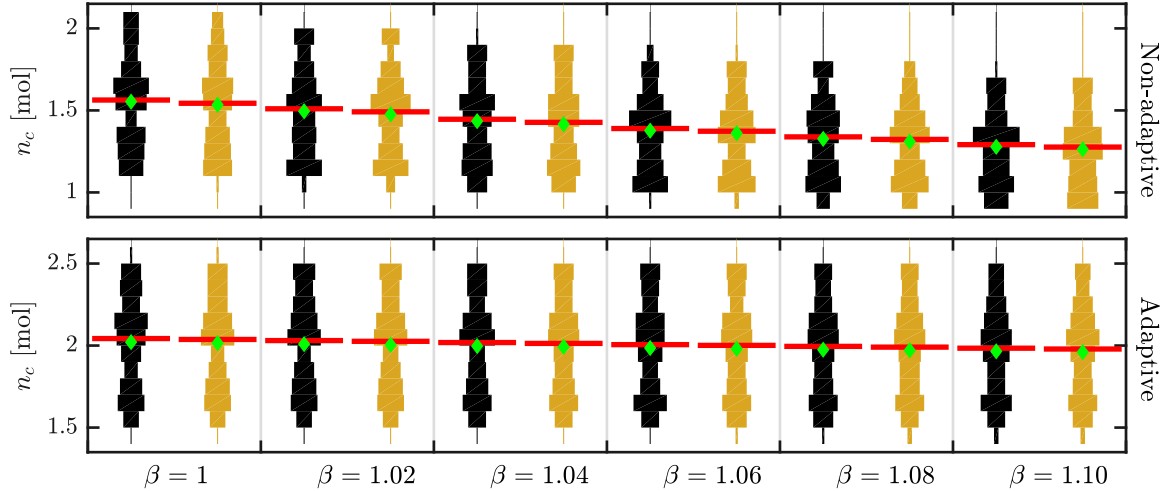

**Figure 12.** Violin plot of the amount of product C produced during the first 0.3 h using MS-SB, A-MS-SB, MS-CB, and A-MS-CB NMPC schemes with different values of the scaling increase factor $(\beta)$ in the absence of measurement noise. ■ represents MS-SB NMPC, A-MS-SB NMPC, and ■ represents MS-CB NMPC, A-MS-CB NMPC. ◆ and ▬ represent the mean and median values.

## 5.2. Case Study 2: Industrial Semi-Batch Polymerization Reactor

The industrial semi-batch polymerization reactor example is taken from [48]. It was provided by BASF SE as a case study for NMPC algorithms in the EU-project [67]. The process flow diagram of the industrial semi-batch reactor under consideration is shown in Figure 13. Monomer A is fed into the reactor, where it combines and produces the polymer P. The reaction is highly exothermic. The reactor

is equipped with an external heat exchanger (EHE) and a cooling jacket to control the temperature of the reactor contents.

The model of the plant was obtained from mass and energy balances. It comprises 8 ordinary differential equations and is given below Equation (26):

$$\dot{m}_W = \dot{m}_F \omega_{W,F}, \tag{26a}$$

$$\dot{m}_A = \dot{m}_F \omega_{A,F} - k_{R1} m_{A,R} - \frac{k_{R2} m_{AWT} m_A}{m_{ges}}, \tag{26b}$$

$$\dot{m}_P = k_{R1} m_{A,R} + \frac{k_{R2} m_{AWT} m_A}{m_{ges}}, \tag{26c}$$

$$\dot{T}_R = \frac{\dot{m}_F c_{p,F}(T_F - T_R)}{c_{p,R} m_{ges}} + \frac{H_R k_{R1} m_{A,R}}{c_{p,R} m_{ges}} - \frac{k_K A(T_R - T_S)}{c_{p,R} m_{ges}} - \frac{\dot{m}_{AWT}(T_R - T_{EK})}{m_{ges}}, \tag{26d}$$

$$\dot{T}_S = \frac{k_K A(T_R - T_S)}{c_{p,S} m_S} - \frac{k_{WS} A(T_S - T_M)}{c_{p,S} m_S}, \tag{26e}$$

$$\dot{T}_M = \frac{\dot{m}_{M,KW}(T_M^{IN} - T_M)}{m_{M,KW}} + \frac{k_{WS} A(T_S - T_M)}{c_{p,W} m_{M,KW}}, \tag{26f}$$

$$\dot{T}_{EK} = \frac{\dot{m}_{AWT}(T_R - T_{EK})}{m_{AWT}} - \frac{\alpha_{exp}(T_{EK} - T_{AWT})}{c_{p,R} m_{AWT}} + \frac{k_{R2} m_A m_{AWT} H_R}{c_{p,R} m_{AWT} m_{ges}}, \tag{26g}$$

$$\dot{T}_{AWT} = \frac{\dot{m}_{AWT,KW}(T_{AWT}^{IN} - T_{AWT})}{m_{AWT,KW}} - \frac{\alpha_{exp}(T_{AWT} - T_{EK})}{c_{p,W} m_{AWT,KW}}, \tag{26h}$$

where

$$U = \frac{m_P}{m_A + m_P}, \tag{26i}$$

$$m_{ges} = m_W + m_A + m_P, \tag{26j}$$

$$k_{R1} = k_0 e^{\frac{-E_a}{R(T_R + 273.15)}} (k_{U1}(1 - U) + k_{U2}U), \tag{26k}$$

$$k_{R2} = k_0 e^{\frac{-E_a}{R(T_{EK} + 273.15)}} (k_{U1}(1 - U) + k_{U2}U), \tag{26l}$$

$$k_K = \frac{m_W k_{WS}}{m_{ges}} + \frac{m_A k_{AS}}{m_{ges}} + \frac{m_P k_{PS}}{m_{ges}}, \tag{26m}$$

$$m_{A,R} = m_A - \frac{m_A m_{AWT}}{m_{ges}}. \tag{26n}$$

The total mass of water, monomer, and polymer present in the reactor and in the external heat exchanger are denoted by $m_W$, $m_A$ and $m_P$. The temperatures of the contents inside the reactor, vessel, and jacket are denoted by $T_R$, $T_S$, and $T_M$. The temperature of the material leaving the external heat exchanger and the coolant inside the external heat exchanger are denoted by $T_{EK}$ and $T_{AWT}$. U gives the ratio of the total mass of the product to the sum of the total mass of the monomer and the product. $m_{ges}$ represents the total mass of the contents of the reactor and the external heat exchanger. $k_{R1}$ and $k_{R2}$ represent the reaction rates of the conversion of the monomer to the polymer inside the reactor and external heat exchanger, respectively. $k_K$ results from an approximation of the heat-transfer coefficient between the reaction mixture and the reactor vessel. The amount of monomer present inside the reactor is given by $m_{A,R}$. The input feed ($\dot{m}_F$), the temperature of the coolant entering the jacket ($T_M^{IN}$) and the temperature of the coolant entering the external heat exchanger ($T_{AWT}^{IN}$) are the control inputs. The bounds on the control inputs are given in Table 6.

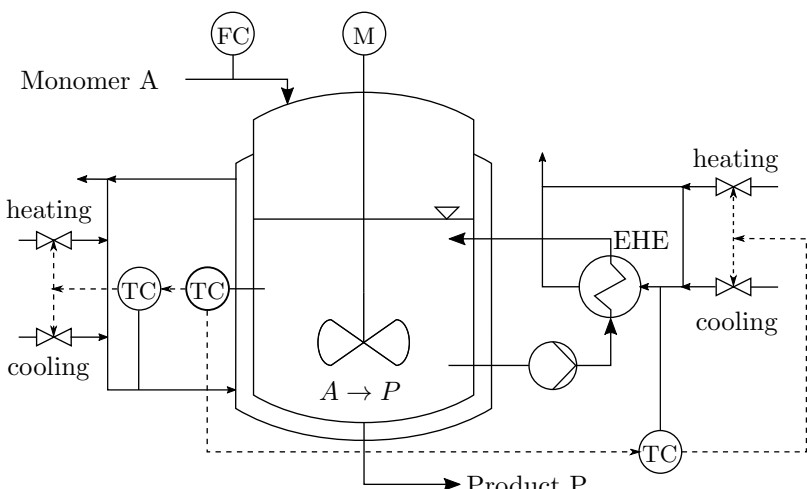

**Figure 13.** Process flow diagram of the industrial polymerization semi-batch reactor.

**Table 6.** Case study 2: Bounds on the control inputs.

| Control | Lower Bound | Upper Bound | Unit |
|---|---|---|---|
| $\dot{m}_F$ | 0 | 30,000 | $\mathrm{kg\,h}^{-1}$ |
| $T_M^{IN}$ | 60 | 100 | °C |
| $T_{AWT}^{IN}$ | 60 | 100 | °C |

### 5.2.1. Formulation of the Optimal Control Problem

The control task is to minimize the batch time while satisfying crucial quality and safety constraints. The temperature of the reactor $T_R$ has to be maintained between 88 °C and 92 °C to obtain the desired product quality. The temperature of the reactor content must not exceed 109 °C in the event of a cooling failure. The maximum temperature of the reactor content that can result in the event of complete cooling failure ($T_{CF}$) is given by the so-called adiabatic temperature Equation (27):

$$T_{CF} = \frac{H_R m_A}{c_{p,R} m_{ges}} + T_R. \tag{27}$$

The total mass of the contents present in the reactor and the external heat exchanger must not exceed 40,879.5 kg. It is assumed the specific reaction rate constant ($k_0$) and the reaction enthalpy ($H_R$) are not known precisely. The parameters are assumed here to be contained in an ellipsoidal confidence region described by their nominal values and the parameter covariance matrix Equation (28):

$$\boldsymbol{d}_0 = \begin{pmatrix} H_R \\ k_0 \end{pmatrix} = \begin{pmatrix} 950\,\frac{\mathrm{kJ}}{\mathrm{kg}} \\ 7 \end{pmatrix}, \boldsymbol{P}_0 = \begin{pmatrix} 36100 & -159.6 \\ -159.6 \times 10^2 & 1.96 \end{pmatrix}. \tag{28}$$

All other parameters present in the model equations are considered to be known and are given in Table 7. All the states ($\boldsymbol{x} = (m_w, m_A, m_P, T_R, T_S, T_M, T_{EK}, T_{AWT})^T$) are considered to be measured directly from the plant with measurement noise of standard deviation $\boldsymbol{\sigma} = (50\,\mathrm{kg}, 50\,\mathrm{kg}, 50\,\mathrm{kg}, 0.1\,°\mathrm{C}, 0.1\,°\mathrm{C}, 0.1\,°\mathrm{C}, 0.1\,°\mathrm{C}, 0.1\,°\mathrm{C})^T$. The initial condition of the states is given as $\boldsymbol{x}_0 = (1000\,\mathrm{kg}, 853\,\mathrm{kg}, 26.5\,\mathrm{kg}, 90\,°\mathrm{C}, 90\,°\mathrm{C}, 90\,°\mathrm{C}, 35\,°\mathrm{C}, 35\,°\mathrm{C})^T$. The sampling time of the controller $t_s$ is chosen as 0.2 h. The batch is completed if the mass of product P produced reaches 20,700 kg. The control vector ($\boldsymbol{u}$) is represented as $(\dot{m}_F, T_M^{IN}, T_{AWT}^{IN})^T$. The MPC optimization problem which is solved at time $t$ ($s^{th}$ iteration) reads as Equation (29):

$$\min_{\boldsymbol{x}_k, \boldsymbol{u}_k, \boldsymbol{\epsilon}_k} \sum_{k=s}^{s+N_p-1} -m_{P,k} + \frac{3.33}{10^3}(\Delta \dot{m}_{F,k})^2 + 2.5(\Delta T_{M,k}^{IN})^2 + 2.5(\Delta T_{AWT,k}^{IN})^2 + 10^5(\epsilon_{k,1}^2 + \epsilon_{k,[2]}^2) + 10^{10}\epsilon_{k,[3]}^2,$$

subject to

$$x_{k+1} = f(x_k, u_k, d_k), \tag{29a}$$

$$88 \le T_{R,k} + \epsilon_{k,[1]} \le 92, \tag{29b}$$

$$\frac{H_R m_A}{c_{p,R} m_{ges}} + T_R + \epsilon_{k,2} \le 109, \tag{29c}$$

$$m_{W,k} + m_{A,k} + m_{P,k} + \epsilon_{k,3} \le 40879.5, \tag{29d}$$

$$-1 \le \epsilon_{k,[1]} \le 1, \tag{29e}$$

$$-1 \le \epsilon_{k,[2]} \le 1, \tag{29f}$$

$$-150 \le \epsilon_{k,[3]} \le 150, \tag{29g}$$

$$\underline{u} \le u_k \le \bar{u}, \tag{29h}$$

$$x_s = x_s^m, \tag{29i}$$

where $\Delta \dot{m}_{F,k} = \dot{m}_{F,k} - \dot{m}_{F,k-1}$, $\Delta T_{M,k}^{IN} = T_{M,k}^{IN} - T_{M,k-1}^{IN}$, $T_{AWT,k}^{IN} - T_{AWT,k-1}^{IN}$, and $\epsilon_k \in \mathbb{R}^{n_c}$. The objective function aims to maximize the number of moles of product P produced along the prediction horizon, and penalizes the control moves and the constraint violations. We assume that maximizing the mass of product P produced along the prediction horizon implicitly minimizes the batch time. The model Equation (26) is discretized using orthogonal collocation on finite elements [68,69] and this leads to Equation (29a). The number of finite elements and the degree of the polynomials are chosen as 4 and 2. The collocation points are obtained using the Radau method [73]. The constraints are given in Equations (29b)–(29d). The constraints are implemented as soft constraints to prevent the MPC optimization from becoming infeasible due to the influence of measurement noise. The maximum constraint violation that can happen is bounded by Equations (29e)–(29g). The control bounds are enforced using Equation (29h), where $\underline{u}_k$ and $\bar{u}_k$ represent the lower and upper bounds on the control and can be obtained from Table 6. The controller is reinitialized at the current plant measurements using Equation (29i), where $x_s^m \in \mathbb{R}^8$ represents the measurement vector obtained at time $t$.

5.2.2. Results

All multi-stage NMPC strategies consider a prediction of horizon $(N_p)$ of length 5 and a robust horizon $(N_r)$ of length 2. All the nodes in the scenario tree of the multi-stage NMPC are equally weighted (i.e., $w_k^i = \frac{1}{N_n}$, s.t. $\sum_{i=1}^{N_n} w_k^i = 1$, where $N_n$ is the number of nodes at time step $k$ in the prediction horizon). The scaling factors $\kappa_s^\star$ and $\kappa_c^\star$ are chosen as 1.75. The scaling increase factor $(\beta)$ is chosen as 1.05. The scaling factors and the scaling increase factors were chosen based on a posterior analysis using simulation studies such that the observed measurements do not invalidate the scaling factors and the NMPC optimization problem is recursively feasible. The value of $\alpha$ was chosen corresponding to a $6\sigma$ confidence level for the adaptive NMPC approaches. The weight on the sigma points $(v)$ chosen to build the covariance matrix of the MS-SB and MS-CB NMPC formulations is given as $(0.2, 0.2, 0.2, 0.2, 0.2)^T$. The nominal values of the uncertain parameters are considered to be the true realizations of the uncertain parameters. The batch is stopped when the measured mass of product P reaches 20,750 kg which guarantees that at least $20,700$ kg of product P $(m_P)$ is present at the batch end. The simulation results obtained when the reactor is controlled using NMPC with the true plant model, the non-adaptive and the adaptive NMPC approaches are shown in Figure 14. The dashed vertical lines indicate the end times of the batches.

**Table 7.** Case study 2: Model parameters.

| Parameter | Description | Value | Unit |
|---|---|---|---|
| $R$ | Universal gas constant | 8.314 | $\text{kJ kmol}^{-1}\text{K}^{-1}$ |
| $c_{p,W}$ | Specific heat capacity of the water (coolant) | 4.2 | $\text{kJ kg}^{-1}\text{K}^{-1}$ |
| $c_{p,S}$ | Specific heat capacity of the steel | 0.47 | $\text{kJ kg}^{-1}\text{K}^{-1}$ |
| $c_{p,F}$ | Specific heat capacity of the feed | 3 | $\text{kJ kg}^{-1}\text{K}^{-1}$ |
| $c_{p,R}$ | Specific heat capacity of the contents inside reactor | 5 | $\text{kJ kg}^{-1}\text{K}^{-1}$ |
| $k_{WS}$ | Heat-transfer coefficient between water and steel | 17,280 | $\text{kJ h}^{-1}\text{m}^{-2}\text{K}^{-1}$ |
| $T_F$ | Feed temperature | 25 | $^\circ\text{C}$ |
| $A$ | Heat-transfer surface of the jacket | 65 | $\text{m}^2$ |
| $m_{M,KW}$ | Mass of coolant in the jacket | 5000 | kg |
| $m_S$ | Mass of the reactor (steel) | 39,000 | kg |
| $m_{AWT}$ | Mass of the product in external heat exchanger | 200 | kg |
| $m_{AWT,KW}$ | Mass of the coolant in external heat exchanger | 1000 | kg |
| $\dot{m}_{M,KW}$ | Mass of the coolant flow into the jacket | 300,000 | $\text{kg h}^{-1}$ |
| $\dot{m}_{AWT,KW}$ | Mass of the coolant flow into the external heat exchanger | 100,000 | $\text{kg h}^{-1}$ |
| $\dot{m}_{AWT}$ | Mass of the product flow into the external heat exchanger | 20,000 | $\text{kg h}^{-1}$ |
| $E_a$ | Activation energy | 8500 | $\text{kJ kmol}^{-1}$ |
| $k_{U1}$ | Reaction parameter 1 | 32 | $-$ |
| $k_{U2}$ | Reaction parameter 2 | 4 | $-$ |
| $\omega_{W,F}$ | Mass fraction of coolant water in the feed | 0.333 | $-$ |
| $\omega_{A,F}$ | Mass fraction of monomer A in the feed | 0.667 | $-$ |
| $k_{AS}$ | Heat-transfer coefficient between monomer and steel | 3600 | $\text{kJ h}^{-1}\text{m}^{-2}\text{K}^{-1}$ |
| $k_{PS}$ | Heat-transfer coefficient between product and steel | 360 | $\text{kJ h}^{-1}\text{m}^{-2}\text{K}^{-1}$ |
| $\alpha_{exp}$ | Experimental coefficient | 3,600,000 | $\text{h}^{-1}$ |

It can be seen that the NMPC with the correct model fully exploits the range of operation and meets the constraints, apart from the effect of measurement errors. The end of the batch is reached at 1.4 h. MS and MS-VA NMPC take into account the presence of uncertainty in the reaction rate constant and in the reaction enthalpy while satisfying the constraint on the reactor temperature for all scenarios in their scenario trees and are shown in Figure 15. The upper and lower bounds on the reactor temperature are active in the predictions of the MS and MS-VA NMPC until the end of the batch. The large range of the uncertain parameters along with the tight bounds on the reactor temperature results in a significant increase in the batch time to 3.2 h. The box over-approximation of the ellipsoidal confidence region adds uncertainty to the parameter estimates which contributes to the increased batch time.

The batch time in the presence of the uncertainties can be reduced using MS-SB and MS-CB NMPC. The sigma points more tightly approximate the uncertainty set. MS-SB and MS-CB NMPC take 2.6 h and 2.4 h to complete the batch. The sensitivity of the states $m_{W,k}$, $m_{A,k}$ $m_{P,k}$ with respect to the uncertain parameters is not zero. This enlarges the box over-approximation of the reachable sets computed using the MS-SB NMPC. This results in an additional performance loss when compared to MS-CB NMPC, since the sensitivity of $m_{W,k} + m_{A,k} + m_{P,k}$ in Equation (29d) with respect to the uncertain parameters is zero.

The batch time can be further reduced significantly using the adaptive NMPC approaches which use the plant measurements to improve the knowledge about the uncertain parameters. A-MS NMPC and A-MS-VA NMPC finish the batch at 1.8 h, close to the optimal batch time if the exact model is used. A-MS-CB and A-MS-SB NMPC require only 1.6 h to finish the batch. There is a 38% and 33% reduction in the batch time when using A-MS-SB and A-MS-CB NMPC over MS-SB and MS-CB NMPC.

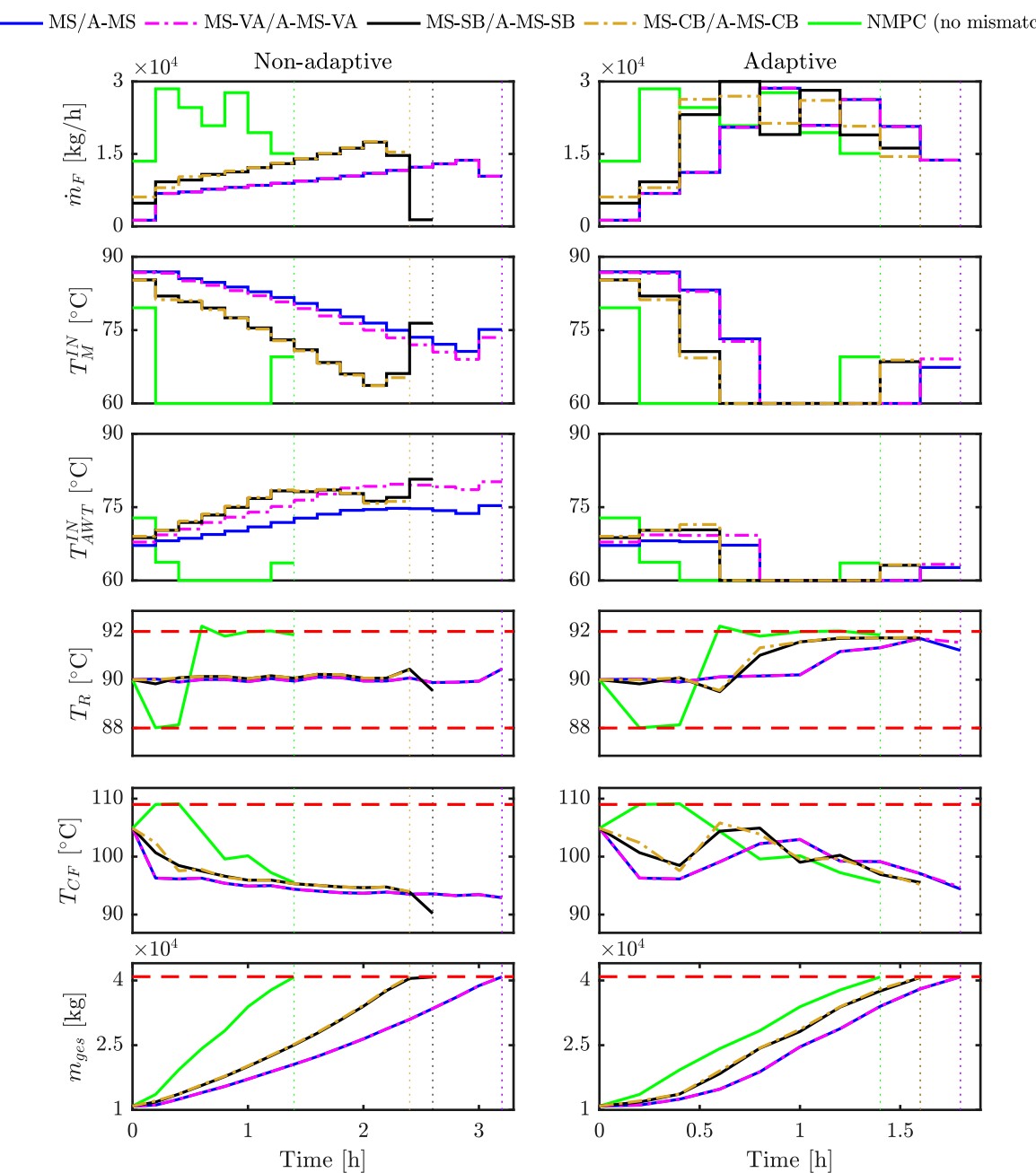

**Figure 14.** Input feed, temperatures of the coolant entering the jacket and entering the external heat exchanger, temperature of the reaction mixture, maximum attainable reactor temperature in the event of a cooling failure, and mass of the contents inside the reactor with their constraints when the reactor is controlled using the different NMPC controllers in the presence of measurement noise. The dotted vertical lines indicate the end times of the batches. Left figure: The plant is controlled using the non-adaptive approaches explained in Section 3, Right figure: The plant is controlled using the adaptive approaches explained in Section 4. NMPC (no mismatch)—the plant model and the simulation model are identical.

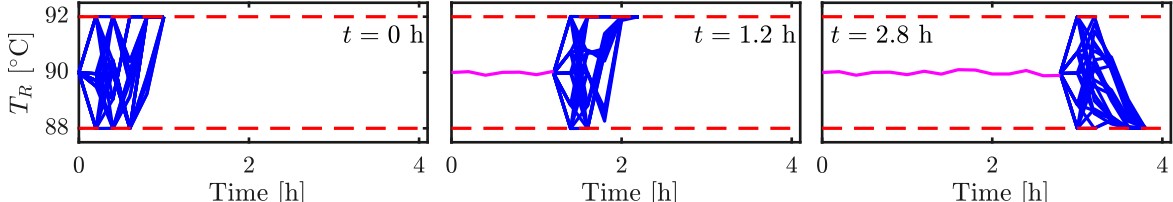

**Figure 15.** Reactor temperature predicted in the scenario tree of the MS-VA NMPC optimization problems solved at $t = 0$ h, $t = 1.2$ h and $t = 2.8$ h. The magenta line (—) indicates the measured reactor temperature and control inputs applied to the plant until time $t$, the blue lines (—) indicate the predicted plant states and control inputs and the dashed red lines (- -) indicate the constraints.

Finally, the performance of the controllers was evaluated for 100 random realizations of the uncertain parameter values within the initial confidence region of the uncertain parameters (marked by black dots in Figure 16a). The results are shown in Figure 16b. The end times of the batches controlled using MS and A-MS NMPC are similar to those of MS-VA and A-MS-VA NMPC because the scenarios generated using the extreme values of the uncertain parameters prevent the controller from feeding more reactants into the reactor and are considered in the scenario tree of both variants. The batch time is considerably reduced using the proposed multi-stage NMPC variants using sigma points when compared to the standard multi-stage NMPC because the uncertainty set is more tightly approximated using the sigma points. MS-CB NMPC results in a better performance when compared to MS-SB NMPC because the performance is restricted by the constraints and their approximation is tighter in MS-CB NMPC. The performance is further improved using the adaptive approaches. A-MS-CB NMPC results in the minimum batch times when compared to other the robust NMPC schemes presented in this paper. The times taken to solve one iteration of the MS, MS-VA, MS-SB, and MS-CB NMPC are 8.89 s, 2.74 s, 12.90 s, and 2.88 s. The times taken to solve one iteration of A-MS, A-MS-VA, A-MS-SB, and A-MS-CB NMPC are 12.47 s, 3.84 s, 28.17 s, and 4.27 s.

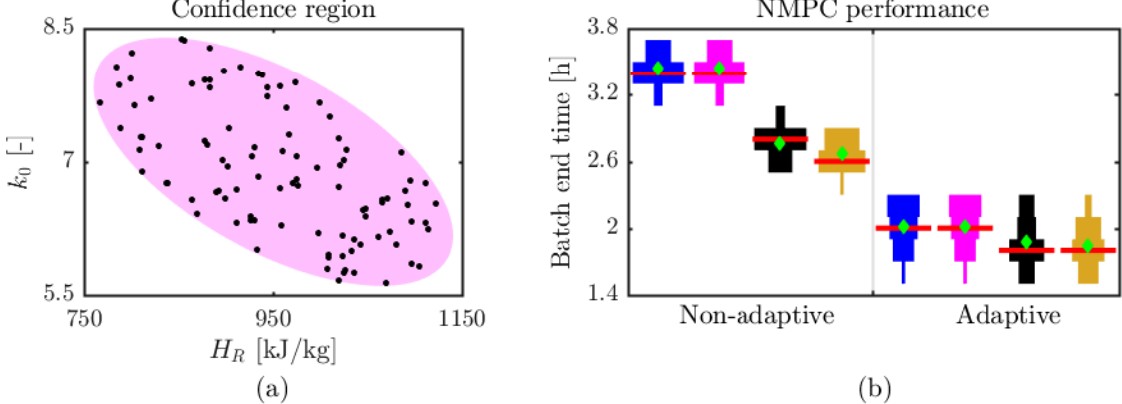

**Figure 16.** (**a**): True plant parameter values chosen within the initial confidence region of the uncertain parameters to evaluate the performance of non-adaptive and adaptive NMPC schemes (**b**): The violin plot of the batch end time obtained when the plant is controlled using non-adaptive and adaptive NMPC schemes in the presence of measurement noise. ■ represents MS NMPC, A-MS NMPC, ■ represents MS-VA NMPC, A-MS-VA NMPC, ■ represents MS-SB NMPC, A-MS-SB NMPC, and ■ represents MS-CB NMPC, A-MS-CB NMPC. ◆ and — represent the mean and median values of the number of moles of product C produced.

## 6. Conclusions

In this paper, we proposed two novel multi-stage NMPC formulations, where the scenario trees are generated using the sigma points, and box over-approximations of the sets of reachable states or constraint function values are computed along the prediction horizon using the unscented

transformation. The proposed scheme tightly over-approximates the uncertainty set and results in a better performance for ellipsoidal uncertainty sets when compared to the traditional multi-stage NMPC where the scenario tree is generated using a box over-approximation. In addition, adaptive approaches are presented where the plant measurements are used to reduce the uncertainty. This significantly reduces the conservatism introduced by the non-adaptive robust NMPC approaches. A simulation study was carried out to analyze the influence of the tuning parameters in the performance of the proposed multi-stage NMPC approaches. An industrial benchmark example is used to show the benefits of the proposed approaches for an example of a realistic size. It confirms the advantages of the new methods. Our future work will focus on dual approaches where a trade-off problem between using excitation signals (probing actions) and performance-oriented control actions is solved. We will also consider improving the performance of robust multi-stage NMPC using sigma points further by over-approximating the reachable sets of the constraint functions and states using an ellipsoidal set instead of a box.

**Supplementary Materials:** The following are available online at http://www.mdpi.com/2227-9717/8/7/851/s1.

**Author Contributions:** Conceptualization, formal analysis, methodology, software, writing—original draft preparation, validation, visualization, S.T.; conceptualization, funding acquisition, methodology, writing—review and editing, validation, R.P.; funding acquisition, supervision, grounding in process engineering, writing—review and detailed editing, validation, S.E.; All authors have read and agreed to the published version of the manuscript.

**Funding:** The authors acknowledge the German Academic Exchange Service (DAAD) and The Ministry of Education, Science, Research and Sport of the Slovak Republic under the Exchange involving project "Reliable and Real-time Feasible Estimation and Control of Chemical Plants". R.P. acknowledges the contribution of the Slovak Research and Development Agency under the project APVV 15-0007 and of the Eur. Commission under the grant 790017 (GuEst). The work of ST was funded by TU Dortmund.

**Acknowledgments:** The authors acknowledge the German Academic Exchange Service (DAAD) and The Ministry of Education, Science, Research and Sport of the Slovak Republic under the Exchange Involving Project "Reliable and Real-time Feasible Estimation and Control of Chemical Plants". RP acknowledges the contribution of the Slovak Research and Development Agency (project APVV 15-0007) and the European Commission (grant 790017).

**Conflicts of Interest:** The authors declare no conflict of interest.

## Abbreviations

The following abbreviations are used in this manuscript:

| | |
|---|---|
| A-MS | Adaptive multi-stage NMPC |
| A-MS-VA | Adaptive multi-stage NMPC based on the vertex over-approximation |
| A-MS-SB | Adaptive multi-stage NMPC based on the box over-approximation of the reachable set of states |
| A-MS-CB | Adaptive multi-stage NMPC based on the box over-approximation of the reachable set of the constraint function |
| EHE | External Heat Exchanger |
| FC | Flow control |
| MPC | Model predictive control |
| NMPC | Nonlinear model predictive control |
| MS | Multi-stage NMPC |
| MS-VA | Multi-stage NMPC based on the vertex over-approximation |
| MS-SB | Multi-stage NMPC based on the box over-approximation of the reachable set of states |
| MS-CB | Multi-stage NMPC based on the box over-approximation of the reachable set of the constraint function |
| TC | Temperature control |

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
