# Peer review of "Robust Multi-Stage Nonlinear Model Predictive Control Using Sigma Points"

_processes, doi:10.3390/pr8070851_

Round 1
Reviewer 1 Report
The paper consitutes a nice contribution to the filed of Nonlinear Model Predictive Control. It is well written and organized. It is noteworthy to mention the systematic approach to present the two case sutdies and the assertive discussion of the results.
Author Response
Please see the attachment below

Reviewer 2 Report
This paper proposes two different methods that the scenario tree of the multi-stage NMPC is built using sigma points for reducing the inevitable loss of performance. It is effective for reducing the performance loss, the method proposed in this paper can be proved is meaningful.
The work of this paper is clear and logical. However, this article still has some deficiencies as follows:
The background of the paper is not clear, there is no detailed description of the overall research ideas.
When observing the experimental result of Figure 9 in Section 5.1, they are not very clear. It will make the reader confused. I think you can change the color or shape in the figure. it might be a better and clearer comparison.
Please proofread it again
Round 2
Reviewer 2 Report
Please check and polish up the presentation again, other issues are well revised in the revision.